# Breaking Latent Prior Bias in Detectors for Generalizable AIGC Image Detection

**Yue Zhou[1], Xinan He[2], Kaiqing Lin[1], Bin Fan[3], Feng Ding[2], Bin Li[1]***

[1]Guangdong Provincial Key Laboratory of Intelligent Information Processing,
Shenzhen Key Laboratory of Media Security,
and SZU-AFS Joint Innovation Center for AI Technology, Shenzhen University
[2]Nanchang University
[3]University of North Texas

## Abstract

Current AIGC detectors often achieve near-perfect accuracy on images produced by the same generator used for training but struggle to generalize to outputs from unseen generators. We trace this failure in part to latent prior bias: detectors learn shortcuts tied to patterns stemming from the initial noise vector rather than learning robust generative artifacts. To address this, we propose **On-Manifold Adversarial Training (OMAT)**: by optimizing the initial latent noise of diffusion models under fixed conditioning, we generate *on-manifold* adversarial examples that remain on the generator's output manifold—unlike pixel-space attacks, which introduce off-manifold perturbations that the generator itself cannot reproduce and that can obscure the true discriminative artifacts. To test against state-of-the-art generative models, we introduce GenImage++, a test-only benchmark of outputs from advanced generators (Flux.1, SD3) with extended prompts and diverse styles. We apply our adversarial-training paradigm to ResNet50 and CLIP baselines and evaluate across existing AIGC forensic benchmarks and recent challenge datasets. Extensive experiments show that adversarially trained detectors significantly improve cross-generator performance without any network redesign. Our findings on latent-prior bias offer valuable insights for future dataset construction and detector evaluation, guiding the development of more robust and generalizable AIGC forensic methodologies. Our dataset and code are publicly available at:
`https://huggingface.co/datasets/Lunahera/genimagepp`

## 1 Introduction

The rapid advancement of generative AI, from early GANs [11] to modern diffusion models [36], has produced increasingly realistic synthesized images, creating a critical need for robust AI-generated content (AIGC) forensic methodologies. While current AIGC detectors perform well on images from their training generator, they often exhibit a significant generalization gap when faced with outputs from unseen generative architectures [48]. This suggests that detectors may learn 'shortcuts'–features specific to the training generator rather than fundamental generative artifacts. Indeed, recent methods achieving better generalization often leverage Stable Diffusion [34] for data augmentation [41, 1, 2], indicating valuable forensic signals within SD data that baseline detectors underutilize.

We hypothesize that a key reason for this shortcut learning is the detector's sensitivity to characteristics tied to the initial latent noise vector, $z_T$. This is related to findings in Xu et al. [42], which showed that the specific 'torch.manual_seed()' for $z_T$ can be identified from generated images, implying

---

*Corresponding author

learnable patterns in the noise sampling process. To investigate this, we conducted white-box attacks by optimizing only $z_T$ (keeping text condition $c$ and the SD generator fixed) against pre-trained CLIP+Linear [33] and ResNet50 [14] AIGC detectors. Our experiments revealed that standard, detectable generated images could consistently be transformed into undetectable ones via these $z_T$ perturbations. Since the generation process remains otherwise unchanged, these resulting adversarial examples lie on the generator's output manifold. We term these **'on-manifold adversarial examples',** distinguishing them from often off-manifold pixel-space attacks. Their successful generation highlights the baseline detectors' reliance on non-robust features linked to $z_T$'s influence.

Building on this, we propose an **on-manifold adversarial training** paradigm. Incorporating these specialized adversarial examples into training significantly improves detector generalization, with our LoRA [18] fine-tuned CLIP-based model achieving state-of-the-art performance on multiple benchmarks. Furthermore, to facilitate rigorous evaluation against contemporary generative models like Black Forest Flux.1 [20] and Stability AI's Stable Diffusion 3 [10], which feature advanced architectures (e.g., DiT [30]) and text encoders (e.g., T5-XXL [6]), we introduce **GenImage++**, a novel test-only benchmark. Our adversarially trained detectors maintain strong performance on this challenging new dataset.

Our contributions are threefold: **(1)** We are the first to systematically expose a vulnerability linked to the detector's sensitivity to initial latent noise variations, identifying this as a significant factor contributing to poor generalization in AIGC detection. **(2)** We are the first to apply on-manifold adversarial training—using only a small set of additional latent-optimized examples—to achieve state-of-the-art generalization in AIGC detection. **(3)** We propose GenImage++, a benchmark incorporating outputs from cutting-edge generators to advance future AIGC forensic evaluation. Collectively, our findings on the detector's latent-space vulnerabilities and the efficacy of on-manifold adversarial training provide valuable insights for future dataset construction and the development of more robust and broadly applicable AIGC forensic methodologies.

## 2 Related Work

The detection of fully synthesized AI-generated content (AIGC) has evolved significantly. Early efforts targeting Generative Adversarial Networks (GANs) [11] often found success with conventional CNNs or by identifying salient frequency-domain artifacts [40]. However, the rise of high-fidelity diffusion models [35, 34] has presented a more formidable challenge, primarily concerning the *generalization* of detectors: models trained on one generator often fail dramatically on outputs from unseen architectures [48, 29]. This generalization gap indicates that detectors frequently learn superficial, generator-specific "shortcuts" rather than fundamental generative fingerprints.

Current state-of-the-art AIGC detectors increasingly leverage powerful pre-trained Vision-Language Models (VLMs) [46], as seen in methods such as UnivFD [29], C2P-CLIP [38], and ReDFD [21]. Similarly, some methods [23, 22, 16]utilize the common-sense knowledge of Multimodal Large Language Models to prevent overfitting, although the generated explanations can be misleading due to a lack of supervision. Another prominent strategy involves utilizing diffusion models themselves, either for reconstruction-based analysis (e.g., DIRE [41], [3]) or extensive training data augmentation (e.g., DRCT [2], PPL [44]), effectively expanding the training manifold. Our work diverges by hypothesizing and exposing a "latent prior bias," where detectors learn non-generalizable shortcuts tied to the initial latent noise $z_T$. We address this by proposing on-manifold adversarial training with **a relatively small, targeted set of $z_T$-optimized examples**. This approach efficiently compels the learning of more robust features, achieving state-of-the-art generalization without relying on massive data augmentation or complex reconstruction pipelines.

## 3 Unveil Latent Prior Bias within AIGC detectors

### 3.1 Standard AIGC Detection and the Generalization Challenge

The fundamental task in AI-generated content (AIGC) forensics is to distinguish between real images, drawn from a distribution $P_R$, and AI-generated images. A standard approach involves training a neural network detector, $D_\phi : \mathcal{X} \to \mathbb{R}$ parameterized by $\phi$, using samples from $P_R$ (labeled as real, $y = 0$) and samples generated by a specific source generative model $G$, drawn from distribution $P_G$

(labeled as fake, $y = 1$). The detector is typically trained by minimizing an expected loss, such as the Binary Cross-Entropy (BCE):

$$\mathcal{L}_{std}(\phi) = \mathbb{E}_{x_R \sim P_R}[\ell(D_\phi(x_R), 0)] + \mathbb{E}_{x_G \sim P_G}[\ell(D_\phi(x_G), 1)], \tag{1}$$

where $\ell(s, y)$ is the BCE loss function. Minimizing $\mathcal{L}_{std}$ enables the detector $D_\phi$ to learn a set of **discriminable features** effective for separating the specific distributions $P_R$ and $P_G$ encountered during training.

However, a critical objective in AIGC forensics is **generalization**: the ability of the detector $D_\phi$, trained primarily on data from generator $G$, to effectively detect images produced by novel, previously unseen generators $G'$ (with distributions $P_{G'}$). Ideally, the features learned by minimizing Eq. (1) should capture fundamental properties indicative of the AI generation process itself, allowing for robust performance across diverse generative models. Yet, empirical evidence often reveals a significant performance degradation when detectors are evaluated on unseen generators, indicating that the learned features may be overly specific to the artifacts of the training generator $G$, a phenomenon often attributed to sample bias and shortcut learning. Addressing this generalization gap is a central challenge we tackle in this work.

## 3.2 Disentangling Process-Inherent Artifacts from Input Conditioning

To develop detectors capable of robust generalization, we must first understand how inputs influence the output of typical generative models and then aim to **disentangle** true forensic features from input-specific characteristics. We examine the structure of modern conditional text-to-image generators, specifically models like Stable Diffusion which often employ Denoising Diffusion Implicit Models (DDIM) [35] for image synthesis. The DDIM process generates a target data sample (latent representation $z_0$) starting from Gaussian noise $z_T \sim \mathcal{N}(0, I)$ via an iterative reverse process over timesteps $t = T, \ldots, 1$. Given the latent state $z_t$ at timestep $t$ and conditioning information $c$ (e.g., text embeddings), a neural network $\epsilon_\theta(z_t, t, c)$ predicts the noise component $\epsilon_{\text{pred}}^{(t)}$ added at the corresponding forward diffusion step. The DDIM sampler then deterministically estimates the previous latent state $z_{t-1}$ using $z_t$ and $\epsilon_{\text{pred}}^{(t)}$:

$$\epsilon_{\text{pred}}^{(t)} = \epsilon_\theta(z_t, t, c), \tag{2}$$

$$z_{t-1} = \sqrt{\bar{\alpha}_{t-1}} \left( \frac{z_t - \sqrt{1 - \bar{\alpha}_t} \epsilon_{\text{pred}}^{(t)}}{\sqrt{\bar{\alpha}_t}} \right) + \sqrt{1 - \bar{\alpha}_{t-1}} \epsilon_{\text{pred}}^{(t)}, \tag{3}$$

where $\bar{\alpha}_t$ represents the noise schedule coefficients. After $T$ steps, the final latent $z_0$ is decoded to the image $x_0 = \text{Dec}(z_0)$ using a VAE decoder Dec.

Crucially, this generation process, $x_0 = \text{SD}(z_T, c)$, involves two primary inputs: the conditioning $c$ and the initial random noise $z_T$. The conditioning $c$ primarily dictates the high-level semantic content and style of the generated image $x_0$. In contrast, the core iterative denoising mechanism (Eq. (3)), the architecture of $\epsilon_\theta$, and the final decoder Dec constitute the fixed machinery of the generator that operates on these inputs.

We posit that the most **generalizable forensic features** ($f_{\text{gen}}$)—those indicative of an image being AI-generated regardless of its specific content or source model—are likely tied to artifacts arising from these intrinsic, input-independent components of the generation pipeline (e.g., the denoising dynamics, network architectures of $\epsilon_\theta$ or Dec, upsampling methods). Since generalizable detection requires recognizing AI generation across diverse content (thus demanding features largely independent of condition $c$), a critical question arises: could detectors, in their attempt to distinguish fakes, inadvertently learn shortcuts from the other primary input, the initial noise $z_T$? If the specific instance or sampling process of $z_T$ imparts any learnable, yet non-generalizable patterns, this could hinder the acquisition of true $f_{gen}$. The finding by Xu et al. [42] that $z_T$ seeds are classifiable from generated images lends plausibility to this concern. Our subsequent methodology is therefore designed to investigate this potential source of shortcut learning.

## 3.3 Exposing Latent Prior Bias via On-Manifold Attack

As established (Section 3.1), standard AIGC detectors often exhibit poor generalization. Building on the concern that detectors might learn shortcuts from the initial noise $z_T$ (Section 3.2), we hypothesize

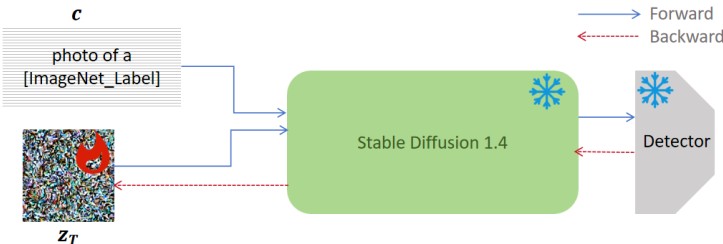

Figure 1: Illustration of our white-box attack methodology. The initial latent noise $z_T$ is iteratively optimized based on gradient feedback from the frozen detector $D_\phi$, while the prompt $c$ and Stable Diffusion generator remain fixed.

that a significant factor contributing to this generalization failure is a phenomenon we term **latent prior bias**. This bias describes the scenario where detectors, trained on images generated from $z_T$ vectors sampled from a prior (e.g., $\mathcal{N}(0, I)$), overfit to image characteristics directly influenced by these $z_T$ instances. Instead of acquiring truly robust generative fingerprints ($f_{gen}$), these detectors learn "shortcuts" by exploiting subtle but learnable patterns tied to the $z_T$-influenced variations. Consequently, the learned decision boundary becomes *biased*, performing well on the seen generator ($P_G$) but poorly on unseen ones ($P_{G'}$).

To directly test this hypothesis and empirically expose the latent prior bias, we designed a white-box **On-Manifold attack** by perturbing the initial latent noise $z_T$. Illustrated in Figure 1, the core objective is to find an optimized latent $z_T^{\mathrm{adv}}$ near an initial random $z_T^{\mathrm{rand}}$ such that the image $x_0^{\mathrm{adv}} = \mathrm{SD}(z_T^{\mathrm{adv}}, c)$, generated with a fixed prompt $c$ and frozen generator SD (Stable Diffusion v1.4), fools a target detector $D_\phi$. This is formulated as an optimization problem where we seek to minimize the detector's confidence that the generated image is fake (i.e., make it appear real, target label $y = 0$). Given $z_T^{(0)} = z_T^{\mathrm{rand}}$, the optimizable latent $z_T^{(k)}$ at iteration $k$ is updated to minimize the following adversarial loss $\mathcal{L}_{\mathrm{adv}}$:

$$\mathcal{L}_{\mathrm{adv}}(z_T^{(k)}) = \ell(D_\phi(\mathrm{SD}(z_T^{(k)}, c)), 0) \tag{4}$$

where $\ell(\cdot, \cdot)$ is the BCEWithLogits loss. The update rule using Stochastic Gradient Descent (SGD) is:

$$z_T^{(k+1)} = z_T^{(k)} - \eta \nabla_{z_T^{(k)}} \mathcal{L}_{\mathrm{adv}}(z_T^{(k)}) \tag{5}$$

where $\eta$ is the learning rate. This iterative optimization (detailed in Algorithm 1, Appendix A.1) continues for a maximum of $K = 100$ steps or until the success condition $\sigma(D_\phi(P(\mathrm{SD}(z_T^{(k)}, c)))) < 0.5$ is met. We applied this attack to GenImage SDv1.4 pre-trained CLIP ViT-L/14 + Linear and ResNet50 detectors in 1. For each of the 1000 unique class labels sourced from the ImageNet-1k dataset [7], we constructed the text prompt $c$ using the template "photo of a [label]". To ensure a substantial and diverse set of adversarial examples for further training, and aiming for a scale comparable to standard test sets, we systematically generated **six successful adversarial examples for each of the 1000 prompts**, using different initial random seeds $s$ for each success. This process yielded our adversarial dataset $X_{\mathrm{adv}}$, comprising $N = 6000$ unique on-manifold adversarial images.

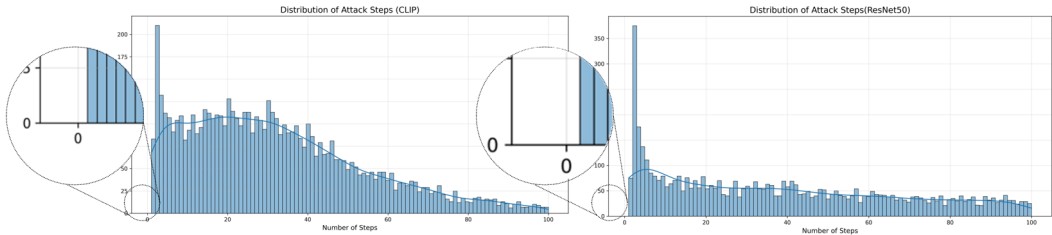

Figure 2: Distribution of optimization steps for successful on-manifold latent attacks targeting CLIP+Linear (left) and ResNet50 (right) detectors. Max optimization steps $K = 100$.

### 3.4 Attack Performance and Evidence of Latent Prior Bias

**Baseline Detectability Confirmed.** Executing the on-manifold latent attack described in Section 3.3 first confirmed the competence of our baseline detectors. As illustrated by the insets in Figure 2, images generated from the unoptimized initial latent noise $z_T^{\mathrm{rand}}$ (i.e., at step $k = 0$ of our attack optimization) were consistently classified as fake by the respective detectors across all 1000 prompts. This establishes that the detectors are effective against typical outputs generated using the standard $z_T$ sampling strategy from the training generator.

**Universal Vulnerability Exposes Latent Prior Bias.** Despite this initial detectability, our subsequent latent optimization revealed a profound vulnerability. For every one of the 1000 prompts, we successfully found multiple initial seeds $s$ for which $z_T^{\mathrm{rand}}$ could be optimized into a $z_T^{\mathrm{adv}}$ that fooled the target detector (i.e., $\sigma(s_{\mathrm{adv}}^{(K')}) < 0.5$). These $z_T^{\mathrm{adv}}$-derived images remained on-manifold, with qualitative inspection (see Appendix Figure 5) confirming their visual coherence and semantic alignment. The consistent success in rendering initially detectable outputs undetectable, achieved solely by perturbing $z_T$, provides strong empirical evidence for the latent prior bias. It demonstrates that the detectors' decisions are critically influenced by learnable, yet non-robust, characteristics tied to the initial latent noise, rather than relying purely on fundamental generative artifacts.

**Attack Efficiency Points to Pervasive Latent Prior Bias.** The dynamics of these successful attacks further characterize this vulnerability. As shown in the main histograms of Figure 2, a significant number of attacks succeeded rapidly (often within $\approx$10 optimization steps). This implies that for many $z_T^{\mathrm{rand}}$, the decision boundary influenced by latent prior bias is very close and easily crossed with minimal $z_T$ perturbation, underscoring the pervasiveness of this bias.

## 4 Mitigating Latent Prior Bias via On-Manifold Adversarial Training

Having established that standard AIGC detectors exhibit a **latent prior bias** by relying on non-robust, $z_T$-influenced shortcut features (Section 3.3), we now introduce our remediation strategy: **On-Manifold Adversarial Training (OMAT)**. OMAT leverages the on-manifold adversarial examples $X_{\mathrm{adv}}$—generated by optimizing $z_T$ to expose these very shortcuts—to retrain the detector.

The core mechanism of OMAT is to alter the detector's learning objective. By augmenting the training data with $X_{\mathrm{adv}}$ (labeled as fake, $y = 1$) and penalizing their misclassification, we render the previously learned $z_T$-influenced shortcuts ineffective for minimizing the training loss. This forces the detector to identify alternative, more robust features. Crucially, since our $X_{\mathrm{adv}}$ are on-manifold (i.e., valid outputs of the generator SD that differ from standard outputs primarily in their $z_T$-derived characteristics, not via off-manifold pixel noise), they retain the intrinsic generative artifacts ($f_{gen}$) common to the generator's process. OMAT thus compels the detector to become sensitive to these $f_{gen}$ as the consistent signals for distinguishing both standard fakes ($P_G$) and our on-manifold adversarial fakes ($X_{\mathrm{adv}}$) from real images ($P_R$).

This retraining process discourages overfitting to superficial $z_T$-influenced characteristics and instead promotes the learning of features more deeply ingrained in the generation process itself. Such features are inherently less dependent on the specific initial $z_T$ and thus more likely to be shared across diverse generative models. This shift—from relying on $z_T$-influenced shortcuts to recognizing fundamental $f_{gen}$—is the key mechanism through which OMAT mitigates latent prior bias and, as demonstrated by our experiments (Section 6), leads to substantial improvements in generalization. This approach aligns with the understanding that robustness to on-manifold adversarial examples is intrinsically linked to generalization [37]. Specific implementation details are in Appendix A.2.

## 5 GenImage++: Evaluating Detectors in the Era of Advanced Generators

While the GenImage dataset [48] provides a valuable baseline, the rapid evolution of generative models necessitates benchmarks that reflect current state-of-the-art capabilities. Recent models, notably FLUX.1 [20] and Stable Diffusion 3 (SD3) [10], signify a leap forward, often employing distinct architectures like Diffusion Transformers (DiT) [30] and significantly more powerful text encoders (e.g., T5-XXL [6]). To provide a more rigorous testbed for generalization against these contemporary capabilities, **we introduce GenImage++**, a novel test-only benchmark dataset. GenImage++ is designed around two core dimensions of advancement, reflecting the "++" in its name:

**Advanced Generative Models:** It incorporates images generated by state-of-the-art models, primarily FLUX.1 and SD3, to directly test detector performance against the latest architectures and their distinct visual signatures.

**Enhanced Prompting Strategies:** It moves beyond simple prompts by including subsets specifically designed to leverage the improved text comprehension of modern generators. This includes images generated from complex, long-form descriptive prompts and a wide array of diverse stylistic prompts.

Figure 3 provides illustrative examples from GenImage++. The benchmark comprises multiple subsets, each targeting specific advancements. These include testing *baseline performance* on standard prompts with Flux and SD3, assessing *long-prompt comprehension*, evaluating robustness to *style diversity* (using Flux, SD3, and comparative models like SDXL and SD1.5), and challenging detectors with high-fidelity *portrait generation*. Full details on its construction, including specific generators, prompt engineering, and subset composition, are provided in Appendix C.

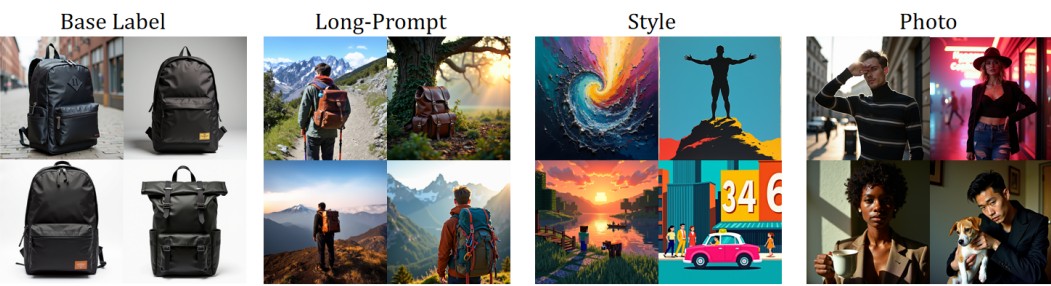

Figure 3: Example images from our proposed GenImage++ benchmark. Columns from left to right: (1) **Base Label** (e.g., "backpack") generated using standard prompts by advanced models; (2) **Long-Prompt** comprehension, where base labels are expanded into complex scenes (e.g., "backpack" in a mountain setting); (3) **Style** diversity, illustrating varied artistic renditions; and (4) **Photo** realism, focusing on high-fidelity human portraits and photorealistic scenes.

# 6 Experiments

## 6.1 Experimental Setup

**Training Paradigm and Our Models.** Our experiments primarily follow the GenImage [48] benchmark paradigm. Unless otherwise specified, all detectors, including our initial baselines and external methods, were trained on the Stable Diffusion v1.4 (SDv1.4) subset of GenImage using real images and corresponding SDv1.4 generated fakes. Our initial detectors are based on ResNet50 [15] and CLIP ViT-L/14 + Linear Head [33]. For adversarial training, these baseline detectors were subsequently fine-tuned using the SDv1.4 training data augmented with $N = 6000$ of our on-manifold adversarial examples ($X_{\text{adv}}$) generated as described in Section 3.3. For CLIP+Linear, adversarial fine-tuning was explored via three strategies: Linear Only, Full Fine-tuning, and LoRA+Linear [18]. These baseline and adversarially trained variants constitute our proposed models.

**Evaluation Datasets and Metrics.** To assess generalization capabilities, we test the trained detectors across a various datasets, including: the original GenImage benchmark; our proposed GenImage++ benchmark (described in Section 5 and Appendix C), which introduces images from advanced generators like Flux and SD3 under challenging conditions; and the Chameleon dataset [43], which comprises diverse AI-generated images collected "in the wild". We assess detector performance using Accuracy (ACC).

**External Baselines for Comparison.** The performance of our models is benchmarked against several external methods. This includes a suite of established standard forensic detectors (e.g., CNNSpot [40], GramNet [25], UniFD [29], SRM [26], AIDE [43]) which analyze image artifacts directly. We also compare against recent state-of-the-art generator-leveraging approaches: DIRE [41], which uses SDv1.4 for reconstruction, and DRCT [2], which employs SDv1.4 for training data augmentation.

**Implementation Details.** Specific hyperparameters for training, LoRA configuration, image preprocessing, dataset compositions, and other relevant implementation choices are provided in Appendix A

Table 1: Generalization performance (Accuracy %) on various GenImage subsets. Models were trained on the SDv1.4 subset. For each column, the best result is marked in **bold** and the second best is underlined. For our adversarially trained models, the change relative to their respective baseline is shown in parentheses (▲ for improvement, ▼ for degradation).

| Model | MidJourney | SDv1.4 | SDv1.5 | ADM | GLIDE | Wukong | VQDM | BigGAN | AVG |
|---|---|---|---|---|---|---|---|---|---|
| *Standard Forensic Methods* | | | | | | | | | |
| Xception[5] | 57.97 | 98.06 | 97.98 | 51.16 | 57.51 | 97.79 | 50.34 | 48.74 | 69.94 |
| CNNSpot[40] | 61.25 | 98.13 | 97.54 | 51.50 | 55.13 | 93.51 | 51.83 | 51.06 | 69.99 |
| F3Net[32] | 52.26 | 99.30 | 99.21 | 49.64 | 50.46 | 98.70 | 45.56 | 49.59 | 68.09 |
| GramNet[25] | 63.00 | 94.19 | 94.22 | 48.69 | 46.19 | 93.79 | 49.20 | 44.71 | 66.75 |
| UniFD[29] | 77.29 | 97.01 | 96.67 | 50.94 | 78.47 | 91.52 | 65.72 | 55.91 | 77.29 |
| NPR[39] | 62.00 | 99.75 | 99.64 | 56.79 | 82.69 | 97.89 | 54.43 | 52.26 | 75.68 |
| SPSL[24] | 56.20 | 99.50 | 99.50 | 51.00 | 67.70 | 98.40 | 49.80 | 63.70 | 73.23 |
| SRM[26] | 54.10 | **99.80** | **99.80** | 49.90 | 52.80 | **99.60** | 50.00 | 51.00 | 69.63 |
| AIDE [43] | 79.38 | 99.74 | 99.76 | 78.54 | 91.82 | 98.65 | 80.26 | 66.89 | 86.88 |
| *Generator-Leveraging Methods* | | | | | | | | | |
| DIRE [41] | 51.11 | 55.07 | 55.31 | 49.93 | 50.02 | 53.71 | 49.87 | 49.85 | 51.86 |
| DRCT/Conv-B [2] | **94.43** | 99.37 | 99.19 | 66.42 | 73.31 | 99.25 | 76.85 | 59.41 | 83.53 |
| DRCT/UniFD [2] | 85.82 | 92.33 | 91.87 | 75.18 | 87.44 | 92.23 | 89.12 | 87.38 | 87.67 |
| *Our Models (Baseline)* | | | | | | | | | |
| ResNet50 | 61.88 | 99.30 | 98.97 | 50.00 | 52.82 | 97.53 | 51.41 | 49.55 | 70.18 |
| CLIP+Linear | 82.11 | 95.89 | 95.43 | 53.82 | 78.89 | 93.30 | 76.61 | 57.53 | 79.20 |
| *Our Models (OMAT)* | | | | | | | | | |
| ResNet50 | 80.49 (+18.61) | 84.11 (-15.19) | 83.46 (-15.51) | 58.13 (+8.13) | 75.32 (+22.50) | 83.22 (-14.31) | 65.58 (+14.17) | 62.76 (+13.21) | 74.13 (+3.95) |
| CLIP Full Fine-tune | 82.18 (+0.07) | 89.70 (-6.19) | 89.68 (-5.75) | 81.45 (+27.63) | 87.65 (+8.76) | 81.94 (-11.36) | 82.28 (+5.67) | 84.99 (+27.46) | 84.98 (+5.78) |
| CLIP Linear Only | 85.77 (+3.66) | 95.33 (-0.56) | 95.05 (-0.38) | 59.40 (+5.58) | 86.79 (+7.90) | 93.60 (+0.30) | 80.65 (+4.04) | 75.95 (+18.42) | 84.07 (+4.87) |
| CLIP+LoRA (Rank 4) | 90.36 (+8.25) | 97.52 (+1.63) | 97.46 (+2.03) | **83.82** (+30.00) | **97.41** (+18.52) | 97.62 (+4.32) | **95.53** (+18.92) | **97.34** (+39.81) | **94.63** (+15.43) |
| CLIP+LoRA (Rank 8) | 89.62 (+7.51) | 95.13 (-0.76) | 94.86 (-0.57) | 79.11 (+25.29) | 94.68 (+15.79) | 94.97 (+1.67) | 90.44 (+13.83) | 91.37 (+33.84) | 91.27 (+12.07) |
| CLIP+LoRA (Rank 16) | 88.57 (+6.46) | 94.82 (-1.07) | 94.32 (-1.11) | 80.98 (+27.16) | 94.27 (+15.38) | 94.69 (+1.39) | 93.82 (+17.21) | 93.22 (+35.69) | 91.84 (+12.64) |

Table 2: **Comparison on the** `Chameleon`**.** Accuracy (%) of different detectors (rows) in detecting real and fake images of `Chameleon`. For each training dataset, the first row indicates the **Acc** evaluated on the `Chameleon` testset, and the second row gives the **Acc** for "fake image / real image" for detailed analysis.

| Training Set | CNNSpot | GramNet | LNP | UnivFD | DIRE | PatchCraft | NPR | AIDE | Our |
|---|---|---|---|---|---|---|---|---|---|
| **SD v1.4** | 60.11 | 60.95 | 55.63 | 55.62 | 59.71 | 56.32 | 58.13 | 62.60 | **66.05** |
| | 8.86/98.63 | 17.65/93.50 | 0.57/97.01 | 74.97/41.09 | 11.86/95.67 | 3.07/96.35 | 2.43/100.00 | 20.33/94.38 | 33.93/90.17 |

## 6.2 Generalization Performance on GenImage

We evaluate the generalization capabilities of our proposed adversarial training strategies and compare them against established baselines on various subsets of the GenImage dataset. Performance, measured by accuracy (%), is reported in Table 1.

Our On-Manifold Adversarial Training strategy consistently enhanced the generalization capabilities of all baseline detectors. Notably, the **OMAT CLIP+LoRA (Rank 4) achieves a state-of-the-art average accuracy of 94.63%** across all GenImage subsets. This marks a +15.43% improvement over its non-adversarially trained baseline (79.20%) and surpasses all other evaluated methods. This superior performance is driven by large gains on challenging OOD subsets (e.g., +30.00% on ADM, +39.81% on BigGAN) while largely maintaining or improving strong performance on in-domain data. These results strongly validate our central hypothesis: on-manifold adversarial training effectively mitigates the identified **latent prior bias** by compelling the detector to learn from challenging latent perturbations—rather than relying on shortcut features tied to initial noise influence—thereby developing more robust, generalizable features that yield consistent high performance across diverse seen and unseen generators.

## 6.3 Performance on the Challenging Chameleon Benchmark

To rigorously evaluate generalization against diverse "in-the-wild" content, we tested our best OMAT model (CLIP+LoRA Rank 4) on the challenging Chameleon dataset [43]. Our model was trained using only the SDv1.4 subset of GenImage, augmented with our on-manifold adversarial examples. Table 2 compares our results against baselines from [43] under the same SDv1.4 training condition. Against this backdrop of established difficulty, our adversarially trained model achieves an average accuracy of 66.05% on Chameleon. This surpasses not only all baselines trained under the comparable SDv1.4 protocol (Table 2) but also notably exceeds the best reported performance (65.77% by AIDE)

Table 3: Performance (Accuracy %) of detectors on our proposed GenImage++ benchmark subsets. All detectors were trained on the SDv1.4 subset of GenImage. Column headers are abbreviated for space: Flux Multi (Flux_multistyle), Flux Photo (Flux_photo), Flux Real (Flux_realistic), SD1.5 Multi (SD1.5_multistyle), SDXL Multi (SDXL_multistyle), SD3 Photo (SD3_photo), SD3 Real (SD3_realistic).

| Model | Flux | Flux Multi | Flux Photo | Flux Real | SD1.5 Multi | SDXL Multi | SD3 | SD3 Photo | SD3 Real | Avg |
|---|---|---|---|---|---|---|---|---|---|---|
| Xception[5] | 36.86 | 10.48 | 4.65 | 5.45 | 97.27 | 20.63 | 38.00 | 5.83 | 15.06 | 26.03 |
| CNNSpot[40] | 37.38 | 6.89 | 8.71 | 5.28 | 84.41 | 34.79 | 47.70 | 7.48 | 25.55 | 28.69 |
| F3Net[32] | 25.18 | 7.79 | 2.83 | 7.90 | 94.15 | 24.01 | 46.67 | 0.84 | 30.28 | 26.63 |
| GramNet[25] | 37.83 | 16.71 | 8.01 | 19.71 | 96.49 | 28.65 | 48.55 | 8.33 | 55.71 | 35.55 |
| NPR[39] | 35.38 | 13.19 | 8.48 | 19.41 | 93.63 | 15.40 | 32.38 | 12.45 | 27.58 | 28.66 |
| SPSL[24] | 67.13 | 16.55 | 43.76 | 25.73 | 71.14 | 17.74 | 44.58 | 16.22 | 29.75 | 36.96 |
| SRM[26] | 8.46 | 2.92 | 0.37 | 1.93 | 96.62 | 6.39 | 9.97 | 0.55 | 4.43 | 14.63 |
| DRCT/UniFD (SDv1)[2] | 71.08 | 63.97 | 46.83 | 62.42 | 99.19 | 64.84 | 72.28 | 70.70 | 73.55 | 69.43 |
| DRCT/Conv-B (SDv1)[2] | 73.02 | 51.91 | 54.72 | 66.40 | 100.00 | 77.19 | 79.10 | 82.93 | 76.58 | 73.54 |
| C2P-CLIP[38] | 0.66 | 0.21 | 0.00 | 0.14 | 41.78 | 18.66 | 1.51 | 0.20 | 0.13 | 7.03 |
| CO-SPY[4] | 94.25 | **95.99** | **99.32** | 98.23 | 85.31 | 58.27 | 75.38 | **92.70** | 92.33 | 87.98 |
| RINE[19] | 93.33 | 67.87 | 77.57 | 84.56 | 97.13 | 94.57 | 97.91 | 90.99 | 91.65 | 88.40 |
| PatchShuffle[45] | 95.83 | 98.15 | 97.30 | **99.50** | 97.82 | 85.65 | 70.67 | 25.30 | 64.28 | 81.61 |
| **Our (CLIP+LoRA R4)** | **96.53** | 92.55 | 97.60 | 97.67 | **100.00** | **99.17** | **98.27** | 90.38 | **98.82** | **96.78** |

for models trained on the **entire GenImage dataset** [43], highlighting the significant generalization imparted by our approach.

## 6.4 Performance on the GenImage++ Benchmark

We further evaluated all detectors trained on the SDv1.4 subset of GenImage against our newly proposed GenImage++ benchmark (described in Section 5). The performance (Accuracy %) on various GenImage++ subsets is detailed in Table 3.

**GenImage++ Exposes Generalization Limits and Architectural Specificity.** The results clearly demonstrate the challenging nature of GenImage++ for detectors trained solely on older SDv1.4 data (Table 3). Most standard forensic methods and even the strong DRCT (SDv1) baseline (69.43% average) struggle significantly on subsets generated by advanced models like Flux and SD3, particularly on their multi-style, photorealistic, and complex prompt variations, with many baseline accuracies falling below 40% or even 20%. This starkly indicates that features learned from SDv1.4 are insufficient for these modern generative capabilities.

**Our OMAT Achieves Robust Generalization on Advanced Models.** In stark contrast, our adversarially trained CLIP+LoRA (Rank 4) model demonstrates vastly superior and remarkably consistent generalization across all subsets of GenImage++, achieving an average accuracy of **96.78%**. It achieves over 90% accuracy on nearly every subset, including those that prove extremely difficult for the baselines. This exceptional performance, indicates that our on-manifold adversarial training successfully forced the learning of more fundamental generative artifacts ($f_{gen}$) that are robust not only across styles within similar architectures but also across entirely different advanced models and their diverse generation modes.

## 6.5 Achieving Generalization through On-Manifold Robustness

To investigate the relationship between generalization and on-manifold attack robustness, we re-attack three CLIP+Linear detector variants, selected based on their varied generalization performance demonstrated in Table 1: (i) the **Baseline** model (poorest generalization), (ii) the **Adv Train Linear** model (moderate generalization), and (iii) the **Adv Train LoRA** (Rank 4) model (best generalization).

Table 4: Robustness of CLIP+Linear detectors against the latent optimization attack after different adversarial training strategies. Attacks were performed on 100 ImageNet labels.

| Metric | Baseline (No Adv Train) | OMAT Linear (Linear Only FT) | OMAT LoRA (LoRA Rank 4 FT) |
|---|---|---|---|
| Success (%) | 75 | 57 | 25 |
| Avg Step | 43.56 | 62.86 | 161.42 |

The results, presented in Table 4, reveal a strong correlation between generalization capability and on-manifold robustness. The Baseline model was most vulnerable. The Adv Train Linear model showed improved robustness, corresponding to its better generalization. Critically, the Adv Train LoRA model, which achieved state-of-the-art generalization, exhibited the highest on-manifold robustness by a significant margin. This clear trend—where models demonstrating superior generalization are also substantially more resilient to on-manifold attacks targeting latent prior bias—provides strong evidence for our hypothesis. It suggests that mitigating this bias through effective on-manifold adversarial training is a key mechanism for developing truly generalizable AIGC detectors.

## 6.6 Comparison with Pixel-Space Adversarial Training

To further contextualize the benefits of our on-manifold latent optimization, we compared its efficacy against adversarial training using examples generated by common $L_\infty$-bounded pixel-space attacks: FGSM [12], PGD [27], and MI-FGSM [8], each with varying perturbation budgets ($\epsilon$). These pixel-space adversarial examples targeted the same baseline CLIP+Linear detector as our latent attacks. The CLIP+LoRA (Rank 4) model was then adversarially trained with these different sets of pixel-space examples. Table 5 presents the average accuracy on the GenImage test subsets.

While adversarial training with all evaluated pixel-space attacks consistently improved generalization over the baseline, these gains are substantially smaller than those from our on-manifold approach. This marked performance difference suggests that our on-manifold adversarial examples, generated by perturbing the initial latent space $z_T$ and processed through the full generative pipeline, are more effective at compelling the detector to learn truly generalizable features $f_{gen}$. Pixel-space attacks, though offering some improvement, seem less adept at correcting the core biases that hinder generalization compared to our on-manifold latent perturbations. This underscores the critical role of the type of adversarial examples in achieving robust generalization for AIGC detection.

Table 5: Impact of adversarial training using different types of adversarial examples on CLIP+LoRA (Rank 4) generalization. All adversarial examples were generated targeting the baseline CLIP+Linear detector trained on SDv1.4.

| Adversarial Training Data Source (Attack Type) | Avg. GenImage Acc (%) |
|---|---|
| Baseline (No Adversarial Training) | 79.19 |
| *RGB-Level Pixel-Space Attacks ($L_\infty$-bounded)* | |
| FGSM ($\epsilon = 0.03$) | 82.11 |
| FGSM ($\epsilon = 0.05$) | 83.18 |
| FGSM ($\epsilon = 0.1$) | 83.49 |
| PGD ($\epsilon = 0.03, \alpha = 0.005, T = 20$) | 82.28 |
| PGD ($\epsilon = 0.05, \alpha = 0.01, T = 20$) | 82.63 |
| PGD ($\epsilon = 0.1, \alpha = 0.02, T = 40$) | 83.11 |
| MI-FGSM ($\epsilon = 0.03, \alpha = 0.005, T = 20$) | 81.76 |
| MI-FGSM ($\epsilon = 0.05, \alpha = 0.01, T = 20$) | 82.61 |
| MI-FGSM ($\epsilon = 0.1, \alpha = 0.015, T = 30$) | 83.16 |
| **On-Manifold Attack** | **94.63** |

## 6.7 Sensitivity to the Adversarial Sample Budget

To understand the impact of the number of adversarial examples on our method's performance, we conduct an ablation study analyzing the sensitivity of OMAT to the size of the adversarial training set. For this experiment, we fine-tune our detector using different proportions of the 6,000 on-manifold adversarial examples we generated, ranging from 0% (the baseline without OMAT) to 100%. We then evaluate the average accuracy on the GenImage benchmark for each configuration.

Table 6: Ablation on the proportion of the 6,000 adversarial examples used for OMAT fine-tuning. We report the average accuracy (%) on the GenImage benchmark.

| % of Adv. Examples Used | 0% | 10% | 25% | 50% | 75% | 100% |
|---|---|---|---|---|---|---|
| Avg. GenImage Acc. (%) | 79.19 | 90.06 | 91.10 | 93.03 | 94.08 | 94.63 |

The results, presented in Table 6, show that OMAT's performance scales gracefully with the adversarial sample budget. Notably, using just 10% of the generated examples (600 images) boosts the

average accuracy from 79.19% to 90.06%, a significant improvement of over 10 percentage points. The performance gains continue to increase with more samples, albeit with diminishing returns, demonstrating that our method is both data-efficient and robust to the exact size of the adversarial set. This confirms that even a modest budget of on-manifold examples is highly effective at mitigating the latent prior bias we identified.

# 7 Conclusion

In this work, we identified **latent prior bias**—a phenomenon where AIGC detectors learn non-generalizable shortcuts tied to the initial latent noise $z_T$—as a key factor limiting their cross-generator performance. We demonstrated this bias empirically using a novel on-manifold attack that optimizes $z_T$ to consistently fool baseline detectors. To mitigate this, we proposed **On-Manifold Adversarial Training (OMAT)**, which leverages these $z_T$-optimized adversarial examples. Our experiments show that OMAT significantly enhances generalization, with our CLIP+LoRA model achieving state-of-the-art performance on established benchmarks and our newly introduced **GenImage++** dataset, which features outputs from advanced generators like FLUX.1 and SD3. Our findings underscore that addressing latent-space vulnerabilities is crucial for developing AIGC detectors that are robust not just to seen generators but also to novel architectures and varied generation conditions. Our work illuminates the path: by systematically identifying and mitigating deep vulnerabilities such as latent prior bias in training data, we can drive the development of forensic tools capable of discerning fundamental, broadly applicable generative fingerprints.

## Acknowledgments and Disclosure of Funding

This work was supported in part by the National Natural Science Foundation of China (Grants U23B2022, U22B2047, 62262041), the Shenzhen R&D Program (Grants JCYJ20250604181211016, SYSPG20241211174032004), and the Jiangxi Provincial Natural Science Foundation (Grant 20232BAB202011).

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

# A Implementation Details

## A.1 Latent Optimization Attack Generation

This section provides detailed hyperparameters and implementation choices for the on-manifold adversarial attack generation process described conceptually in Section 3.3 and Algorithm 1.

**Objective Recap.** The goal was to iteratively optimize an initial latent noise vector $z_T^{\text{rand}}$ into an adversarial vector $z_T^{\text{adv}}$ such that the generated image $x_0^{\text{adv}} = \text{SD}(z_T^{\text{adv}}, c)$ fooled a target detector $D_\phi$ ($\sigma(D_\phi(P(x_0^{\text{adv}}))) < 0.5$), while keeping the generator SD and prompt $c$ fixed.

---

**Algorithm 1** Single Attempt: On-Manifold Latent Noise Optimization Attack

---

**Require:** Frozen detector $D_\phi$, Frozen generator SD, Text prompt $c$, Seed $s$
**Require:** Max steps $K = 100$, Learning rate $\eta = 1 \times 10^{-3}$, Success threshold $\tau = 0.5$
**Require:** Preprocessing function $P(\cdot)$, Loss function $\ell(s, y) = \text{BCEWithLogits}(s, y)$
  1: Set random seed using $s$
  2: Initialize $z_T^{(0)} \sim \mathcal{N}(0, I)$; $z_T^{\text{opt}} \leftarrow z_T^{(0)}$
  3: $z_T^{\text{adv}} \leftarrow$ None
  4: **for** $k = 0$ to $K - 1$ **do**
  5:     $x_0^{(k)} \leftarrow \text{SD}(z_T^{\text{opt}}, c)$                            ▷ Generate image (35 inference steps)
  6:     $s_{\text{adv}}^{(k)} \leftarrow D_\phi(P(x_0^{(k)}))$ ▷ Get detector logit ($P(\cdot)$ includes postprocess & Kornia transforms)
  7:     $L_{\text{adv}}^{(k)} \leftarrow \ell(s_{\text{adv}}^{(k)}, 0)$
  8:     **if** $\sigma(s_{\text{adv}}^{(k)}) < \tau$ **then**
  9:         $z_T^{\text{adv}} \leftarrow z_T^{\text{opt}}.\text{detach}()$
 10:         **break**
 11:     **end if**
 12:     $g \leftarrow \nabla_{z_T^{\text{opt}}} L_{\text{adv}}^{(k)}$
 13:     $z_T^{\text{opt}} \leftarrow z_T^{\text{opt}} - \eta g$                                                     ▷ SGD update
 14: **end for**
**Ensure:** Resulting $z_T^{\text{adv}}$ if attack succeeded, otherwise None.

---

**Generator and Detector Setup.** We used Stable Diffusion v1.4 (CompVis/stable-diffusion-v1-4 via Hugging Face Diffusers) as the frozen generator SD. Attacks targeted our baseline ResNet50 [14] and CLIP+Linear [33] detectors, which were pre-trained on the SDv1.4 GenImage subset.

**Input Initialization.** For each attack, a text prompt $c$ was formatted as "photo of [ImageNet Label]", using labels from ImageNet-1k [7]. The initial latent noise $z_T^{(0)}$ was sampled from $\mathcal{N}(0, I)$ with shape $(1, 4, 64, 64)$ and `dtype=torch.float16`. Reproducibility for each attempt was ensured using `torch.cuda.manual_seed(seed_count)`. This initial tensor $z_T^{(0)}$ was wrapped in `torch.nn.Parameter` to enable gradient computation and designated as the optimizable variable $z_T^{\text{opt}}$.

**Optimization Loop Details.** The optimization proceeded for a maximum of $K = 100$ steps. In each step $k$:

- *Generation:* The current latent $z_T^{(k)}$ was input to the frozen SD pipeline (`pipe`) with `output_type='latent'` and `num_inference_steps=35`. The resulting UNet output latents were scaled (`/ pipe.vae.config.scaling_factor`) and decoded using the frozen VAE decoder (`pipe.vae.decode`) into an intermediate image tensor, which was then cast to `torch.float32`.

- *Preprocessing $P(\cdot)$:* The decoded image tensor underwent two stages of preprocessing. First, a custom `postprocess` function was applied: the image was denormalized to $[0, 1]$ (via `* 0.5 + 0.5` and clamping), scaled to $[0, 255]$, subjected to a simulated precision loss mimicking float-to-uint8-to-float conversion using the operation `image += image.round().detach() - image.detach()`, and finally rescaled to $[0, 1]$. This step aimed to simulate realistic image saving/loading artifacts. Sec-

ond, standard detector-specific preprocessing (`discriminator_preprocess`) was applied using the Kornia library [9] to ensure differentiability. This involved resizing the image tensor to $224 \times 224$ pixels (`K.Resize(..., align_corners=False, antialias=True)` followed by `K.CenterCrop(224)`) and normalizing it using the standard CLIP statistics (`K.Normalize(mean=[0.4814..., 0.4578..., 0.4082...], std=[0.2686..., 0.2613..., 0.2757...])`).

- *Loss & Optimization:* The preprocessed image was passed through the frozen detector $D_\phi$ to obtain the logit $s_{\text{adv}}^{(k)}$. The adversarial loss $L_{\text{adv}}^{(k)}$ was calculated using `torch.nn.BCEWithLogitsLoss` with a target label of 0. Gradients $\nabla_{z_T^{(k)}} L_{\text{adv}}^{(k)}$ were computed via backpropagation through the entire differentiable path (detector, Kornia preprocessing, custom postprocessing, VAE decoder, diffusion steps). The latent $z_T^{\text{opt}}$ was updated using `torch.optim.SGD` with a learning rate $\eta = 1 \times 10^{-3}$.

**Termination and Output.** The optimization loop terminated if the success condition $\sigma(s_{\text{adv}}^{(k)}) < 0.5$ was met or after $K = 100$ steps. If successful at step $K'$, the resulting latent $z_T^{\text{adv}} = z_T^{(K')}$ was used to generate the final adversarial image $x_0^{\text{adv}} = \text{SD}(z_T^{\text{adv}}, c)$, which was saved as a PNG file. We iterated through seeds for each prompt until one successful adversarial example was generated, collecting a total of $N = 6000$ examples for the $X_{\text{adv}}$ dataset.

**Environment.** Experiments were conducted using PyTorch 2.5.0dev20240712, Diffusers 0.33.0.dev0, Kornia 0.8.0, and single NVIDIA A100 GPU with CUDA 12.4.

## A.2  Adversarial Training / Fine-tuning Details

This section details the process used to fine-tune the baseline detectors using the generated on-manifold adversarial examples ($X_{\text{adv}}$).

**Objective.** The goal was to improve the robustness and generalization of the initial baseline detectors (ResNet50 and CLIP+Linear, pre-trained on SDv1.4) by further training them on a dataset augmented with the $N = 6000$ adversarial examples $X_{\text{adv}}$ generated as described in Section A.1.

**Dataset Composition.** The training dataset was formed by combining the original SDv1.4 training set from GenImage [48] (real images labeled $y = 0$, SDv1.4 fakes labeled $y = 1$) with the full set of $N = 6000$ adversarial examples $X_{\text{adv}}$ (also labeled as fake, $y = 1$). Standard PyTorch `Dataset` and `DataLoader` classes were used, with the adversarial examples loaded via a separate dataset instance and iterated through alongside the standard data within each epoch, as shown in the `train_epoch` function structure. The combined dataset was shuffled for training.

**Common Training Parameters.** Key parameters applied across most fine-tuning runs include:

- **Optimizer:** AdamW.
- **Weight Decay:** $1 \times 10^{-4}$.
- **Loss Function:** Binary Cross-Entropy with Logits (`torch.nn.BCEWithLogitsLoss`).
- **Epochs:** 20.
- **Adversarial Weighting:** The loss contribution from adversarial samples ($x \in X_{\text{adv}}$) was dynamically weighted using $\lambda_{\text{adv}} = \min(1.0 + 0.2 \times \text{epoch}, 3.0)$, linearly increasing the emphasis on adversarial samples from 1.2 up to 3.0 over the first 10 epochs, then capping at 3.0.
- **Checkpointing:** Model checkpoints were saved after each epoch. The "best model" was selected based on a combined score: $0.6 \times \text{Val Acc} + 0.4 \times \text{Adv Sample Acc}$, where Avg Val Acc is the accuracy on SDv1.4 validation set, and Adv Sample Acc is the accuracy on the $X_{\text{adv}}$ set itself evaluated after each epoch.

**Model-Specific Fine-tuning Strategies.**

- **ResNet50 (Full Fine-tuning):** The entire baseline ResNet50 detector was fine-tuned end-to-end using the augmented dataset. The learning rate was set to $1 \times 10^{-4}$ and batch size to 128. Input images were preprocessed using Kornia [9] for resizing/cropping to $224 \times 224$ and normalization with standard ImageNet statistics (mean=[0.485, 0.456, 0.406], std=[0.229, 0.224, 0.225]).

- **CLIP+Linear (Full Fine-tuning):** The entire CLIP ViT-L/14 backbone and the linear head were fine-tuned end-to-end using the augmented dataset. Learning rate and batch size were likely similar to the ResNet50 setup. Preprocessing used CLIP's statistics and $224 \times 224$ resolution (Section A.1).
- **CLIP+Linear (Linear Only):** Only the final linear classification head was fine-tuned using the augmented dataset. The CLIP backbone remained frozen. Learning rate and batch size were likely similar to the ResNet50/Full FT setup [Confirm LR/BS if different]. Preprocessing used CLIP's statistics.
- **CLIP+Linear (LoRA+Linear):** LoRA [18] was applied to the CLIP ViT-L/14 backbone while simultaneously fine-tuning the linear head. We used the PEFT library with `LoraConfig` set for `TaskType.FEATURE_EXTRACTION`. Experiments were run with LoRA ranks $r \in \{4, 8, 16\}$. The LoRA alpha was set to $\alpha = r \times 2$, dropout was $0.1$, and target modules were `"q_proj"`, `"v_proj"`, `"k_proj"`, `"out_proj"`. The learning rate for this strategy was $2 \times 10^{-4}$ and batch size was 32. Preprocessing used CLIP's statistics.

## B   Additional Results for On-Manifold Latent Attack

This section provides further analysis of our proposed on-manifold latent optimization attack, focusing on the quality of the generated adversarial images and characteristics of the optimized latent noise vectors.

### B.1   Qualitative Analysis and Image Fidelity

A key characteristic of our on-manifold attack, which perturbs the initial latent noise $z_T$ rather than directly manipulating pixel values of a rendered image, is its ability to maintain high visual fidelity and semantic coherence with the input prompt $c$. Unlike traditional pixel-space attacks where perturbations might appear as noise, our method guides the generator SD to produce images that are still within its learned manifold but possess subtle characteristics that fool the detector.

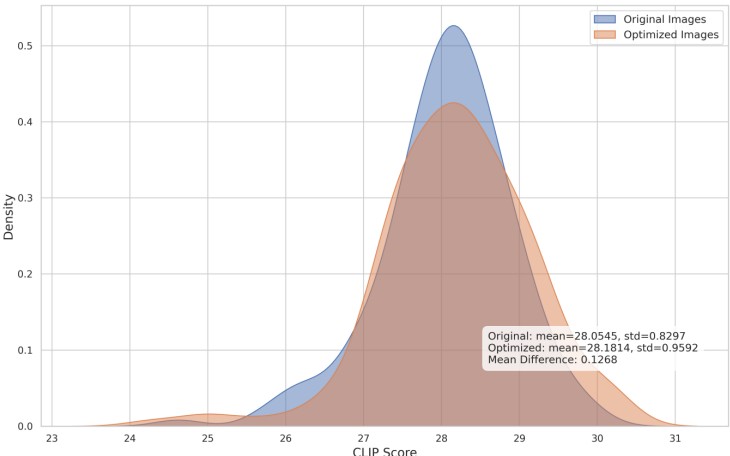

Figure 4: Distribution of CLIP scores for images generated from original random latent noise ($z_T^{\mathrm{rand}}$) versus optimized adversarial latent noise ($z_T^{\mathrm{adv}}$) for the prompt "photo of a cat". The close similarity in distributions and mean scores (Original: mean=28.05, std=0.83; Optimized: mean=28.18, std=0.96; Mean Difference: 0.13, based on [Number] samples) indicates that the latent optimization process preserves image fidelity and prompt alignment.

To quantitatively assess image fidelity and alignment with the prompt, we computed CLIP scores [33, 17] between the generated images and their corresponding text prompts. We compared the distributions of CLIP scores for images generated from original random latent noise $z_T^{\mathrm{rand}}$ (before optimization, detectable by the baseline detector) with those generated from the optimized adversarial latent noise $z_T^{\mathrm{adv}}$ (after successful attack, fooling the baseline detector). Figure 4 illustrates the CLIP score distributions for the prompt "photo of a [ImageNet_Label]", based on 6000 pairs of

original and optimized images. The distribution for optimized (adversarial) images largely overlaps with that of the original images. The mean CLIP score for original images was 28.05 (std=0.83), while for the optimized adversarial images it was 28.18 (std=0.96), resulting in a mean difference of only 0.13. This minimal difference and substantial overlap indicate no significant degradation in perceived image quality or prompt alignment due to the latent optimization process. Similar trends were observed across other prompts.

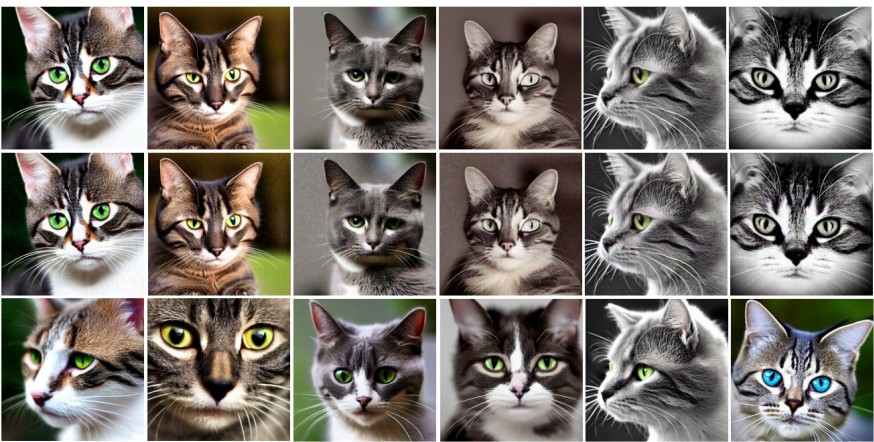

Figure 5: Comparison of adversarial examples for the prompt "photo of a cat" across multiple initial generations (columns). Rows from top to bottom: Original generated images, pixel-space attack samples (FGSM), and our on-manifold adversarial samples. Our on-manifold attacks fool the detector while preserving intrinsic generative artifacts, unlike pixel-space attacks that often introduce disruptive noise.

Figure 8 provides a qualitative comparison across multiple ImageNet labels, showcasing images generated using various $L_\infty$-bounded pixel-space attacks (FGSM, PGD, MI-FGSM with different $\epsilon$ values) alongside examples from our on-manifold latent optimization attack. As can be observed, the pixel-space attacks, particularly at higher $\epsilon$ values (e.g., $\epsilon = 0.1$), often introduce noticeable, somewhat unstructured noise patterns across the image. While these perturbations are bounded and may be "imperceptible" in the sense of not drastically changing the overall scene, their texture can appear artificial or noisy. In contrast, the images generated by our on-manifold attack (rightmost column for each label) maintain a high degree of visual realism and coherence consistent with the output of the Stable Diffusion generator. The differences from an original, non-adversarial generation (not explicitly shown side-by-side here, but understood to be visually similar to the on-manifold attack output) are typically semantic or structural (e.g., slight changes in pose, background detail, lighting, or object features), rather than additive noise. This visual evidence further supports the claim that our latent optimization produces on-manifold adversarial examples that preserve image quality while effectively fooling the detector.

## B.2 Analysis of Optimized Latent Noise Vectors

To investigate whether the optimized adversarial latent vectors $z_T^{\text{adv}}$ occupy a distinct region or cluster within the latent space compared to the initial random noise vectors $z_T^{\text{rand}}$, we conducted a focused experiment. Using the prompt "a photo of a cat", we performed our latent optimization attack targeting the baseline CLIP+Linear detector for 200 unique random seeds ('torch.manual_seed(0)' to 'torch.manual_seed(199)'). Each attack was limited to a maximum of $K = 50$ optimization steps.

Out of the 200 attempts, 164 seeds resulted in successful adversarial latents $z_T^{\text{adv}}$ within the 50-step budget. We then collected all 200 initial random latents $z_T^{\text{rand}}$ (Original Latents) and the 164 corresponding successful adversarial latents $z_T^{\text{adv}}$ (Optimized Latents). To visualize their distribution, we applied t-SNE to project these high-dimensional latent vectors (shape $1 \times 4 \times 64 \times 64$, flattened) into a 3D space.

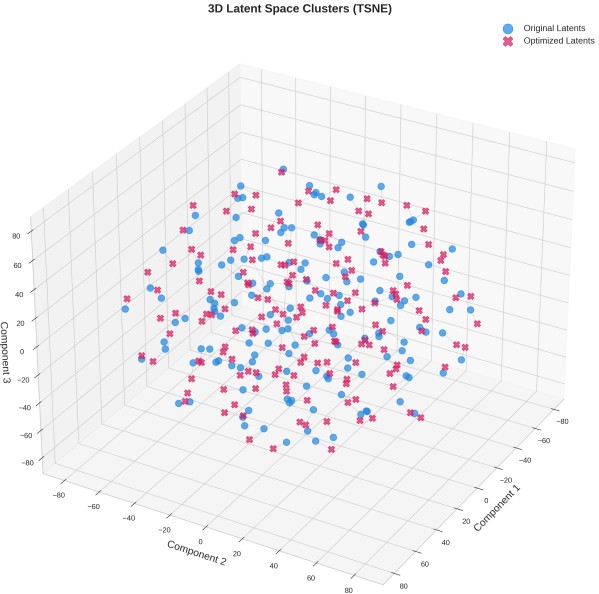

Figure 6: 3D t-SNE visualization of initial random latent vectors ($z_T^{\mathrm{rand}}$, blue circles) and successfully optimized adversarial latent vectors ($z_T^{\mathrm{adv}}$, pink 'x' markers) for the prompt "a photo of a cat". The lack of distinct clustering indicates that optimized latents are not confined to a specific sub-region but are interspersed among the original random latents.

The resulting visualization is shown in Figure 6. As can be seen, the Optimized Latents (marked with 'x') do not form distinct, well-separated clusters from the Original Latents (marked with 'o'). Instead, the optimized adversarial latents appear to be intermingled with and distributed similarly to the initial random latents. This suggests that successful adversarial perturbations do not necessarily push the latents into a far-off, specific region of the latent space. Rather, they represent relatively small deviations from the initial random points, sufficient to cross the detector's decision boundary.

This observation is consistent with the notion that the detector's decision boundary is brittle and that many points on the generator's manifold are close to "adversarial pockets" reachable via minor latent adjustments. The lack of clear clustering for $z_T^{\mathrm{adv}}$ further implies that the vulnerability exploited is not tied to a single, easily characterizable sub-distribution of "bad" or "outlier" initial latents. Instead, it suggests that the **standard detector fails to robustly interpret subtle variations across a broad range of typical initial latents** $z_T^{\mathrm{rand}}$. The issue is not that only certain types of initial latents are problematic, but rather that the detector's learned features are sensitive to small, targeted changes originating from many diverse starting points in the latent space. This reinforces the idea that the detector relies on non-robust "shortcut" features that can be easily masked or manipulated through these nuanced latent perturbations.

## C   The GenImage++ Benchmark Dataset Construction

This appendix details the motivation, design, and construction of our GenImage++ benchmark dataset.

### C.1   Motivation and Design Goals

While the GenImage dataset [48] provides a valuable baseline, its constituent models predate significant recent advancements. To offer a more contemporary and challenging evaluation, we developed GenImage++. Our design addresses two primary dimensions of progress in generative AI: (1) **Advanced Generative Models**, incorporating outputs from state-of-the-art architectures like FLUX.1 and Stable Diffusion 3 (SD3), which utilize Diffusion Transformers (DiT) and powerful text encoders (e.g., T5-XXL); and (2) **Enhanced Prompting Strategies**, leveraging the improved prompt adherence

and diverse generation capabilities of these modern models. GenImage++ is a **test-only** benchmark designed to rigorously assess AIGC detector generalization against these new capabilities.

## C.2  Generative Models Used

The following generative models were used, sourced from Hugging Face Transformers and Diffusers libraries:

- **FLUX.1-dev:** `black-forest-labs/FLUX.1-dev` [20].
- **Stable Diffusion 3 Medium (SD3):** `stabilityai/stable-diffusion-3-medium` [10].
- **Stable Diffusion XL (SDXL):** `stabilityai/stable-diffusion-xl-base-1.0` [31].
- **Stable Diffusion v1.5 (SD1.5):** `runwayml/stable-diffusion-v1-5` [34].

For FLUX.1-dev and SD3, images were generated at $1024 \times 1024$ resolution using 30 inference steps, a guidance scale of 3.5, and a maximum sequence length of 512 tokens, unless specified otherwise. Parameters for SDXL and SD1.5 followed common practices for those models.

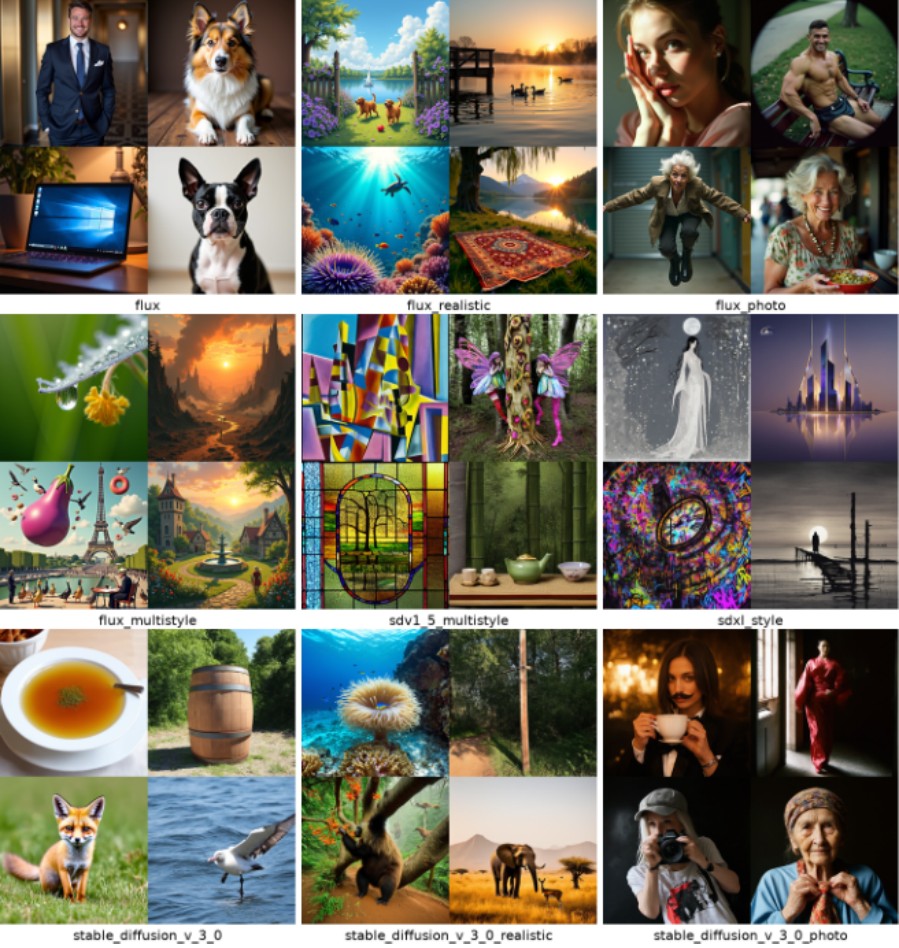

Figure 7:  Sample images from multiple subsets of the GenImage++ dataset. Each block (from left to right, top to bottom) shows examples generated by different models and style presets: (a) Flux, (b) Flux realistic, (c) Flux photo, (d) Flux multistyle, (e) SDv1.5 multistyle, (f) SDXL style, (g) Stable Diffusion v3.0, (h) Stable Diffusion v3.0 realistic, and (i) Stable Diffusion v3.0 photo. This figure illustrates the visual diversity across models and styling configurations in our dataset.

### C.3 Subset Construction and Prompt Engineering

GenImage++ comprises several subsets, each with distinct prompt engineering strategies, leveraging Meta-Llama-3.1-8B-Instruct (`meta-llama/Llama-3.1-8B-Instruct`) [13] for prompt expansion where noted. The scale of each generated subset is detailed in Table 7.

Table 7: Number of images in each subset of the GenImage++ benchmark.

| Subset | Flux | Flux_multi | Flux_photo | Flux_real | SD1.5_multi | SDXL_multi | SD3 | SD3_photo | SD3_real |
|---|---|---|---|---|---|---|---|---|---|
| Images | 6k | 18.3k | 30k | 6k | 18.3k | 18.3k | 6k | 30k | 6k |

**Base Subsets (Flux, SD3):**

- *Target:* Baseline performance on FLUX.1 and SD3.
- *Prompting:* Following the GenImage protocol, "photo of a [ImageNet Label]" using labels from ImageNet-1k [7].
- *Scale:* 6000 images per model (see Table 7).

**Realistic Long-Prompt Subsets (Flux_realistic, SD3_realistic):**

- *Target:* Long-prompt comprehension and detailed scene generation.
- *Prompting:* ImageNet labels were expanded by Llama-3.1 using the system prompt: "You are now a text-to-image prompt generator. For the given ImageNet class labels, integrate them into more harmonious and detailed scenes. Provide the prompt directly without any additional output." The ImageNet label was then provided as the user message.
- *Scale:* As per Table 7.
- *Example:* **Label: Goldfish** -> "In a serene Zen garden, a majestic goldfish swims majestically in a tranquil pool of crystal-clear water, surrounded by intricately raked moss and stone formations that reflect the warm sunlight filtering through a waterfall in the background."

  **Label: White Shark** -> "A colossal great white shark swimming effortlessly through crystal-clear, turquoise waters, surrounded by a vibrant coral reef, teeming with an array of tropical fish, including iridescent blue damselfish and shimmering yellow tang, while a sunny sky with puffy white clouds reflects off the rippling ocean surface.",

**Multi-Style Subsets (Flux_multistyle, SDXL_multistyle, SD1.5_multistyle):**

- *Target:* Style diversity and adherence.
- *Prompting:* Based on 183 style templates from the ComfyUI_MileHighStyler repository[2]. For each style template (which contains a placeholder for a scene), Llama-3.1 generated a fitting scene description using the system prompt: "I have a style description that contains a placeholder for a specific scene, marked as {{prompt}}. Please provide a detailed and vivid description to fill in this placeholder. Only return a single scene description for {{prompt}}, without including any additional text or references to the style or the prompt itself. Style: [original style template from JSON file]".
- *Scale:* 100 images per style for each of the 183 styles, totaling 18,300 images per model (Flux, SDXL, SD1.5). Style compatibility for FLUX.1 was referenced from a community-maintained list[3].
- *Example:* "name": "sai-origami", "prompt": "origami style In a tranquil morning mist, delicate petals unfold like a gentle lover's whisper, as if the very essence of cherry blossoms had been distilled into this exquisite origami piece. Soft pink hues dance across the intricate pleats, evoking the tender promise of spring's awakening. The delicate flower, masterfully crafted from layers of pleated paper, rises from the serene landscape like a whispering prayer. Each fold and crease has been carefully crafted to evoke the subtle nuances of nature's beauty, its beauty so captivating that it appears almost lifelike. Its central axis, a slender stem, supports the delicate blossom, guiding it upwards with an elegance that

---

[2]`https://github.com/TripleHeadedMonkey/ComfyUI_MileHighStyler/blob/main/sdxl_styles_twri.json`

[3]`https://enragedantelope.github.io/Styles-FluxDev/`

belies the complexity of its creation. Around it, the gentle curves of the paper seem to blend seamlessly with the misty atmosphere, creating an ethereal balance between the organic and the man-made. The overall composition is a testament to the timeless beauty of origami art, where the simplicity of a single piece can evoke a profound sense of serenity and wonder, inviting the viewer to pause and appreciate the intricate beauty that unfolds before them. . paper art, pleated paper, folded, origami art, pleats, cut and fold, centered composition" ,

"name": "artstyle-graffiti", "prompt": "graffiti style The city's concrete heart beats stronger under the vibrant glow of a street art haven. Amidst a maze of alleys, where the city's underbelly roars to life at dusk, a towering mural sprawls across a worn brick wall. "Phoenix Rising" - the mural's bold title is emblazoned across the top in fiery red and orange hues, its letters splattered with an explosive mixture of colors that dance like flames. Below, a majestic bird bursts forth from the ashes, its wings outstretched, radiating a kaleidoscope of colors - sapphire, emerald, and sunshine yellow. In the background, a cityscape is reduced to smoldering embers, the remnants of a ravaged metropolis that has been reborn through the unrelenting power of art. Skyscrapers, reduced to skeletal frames, pierce the night sky like splintered shards of glass. Smoke billows from the rooftops, but amidst the destruction, a phoenix rises. Its eyes, like two glittering opals, seem to shine with an inner light, as if the very essence of rebirth has been distilled within its being. The mural's tag, signed in bold, sweeping script as "Kaos", runs across the bottom of the image, a rebellious declaration of the artist's intent: to shatter the conventions of urban decay and bring forth a radiant new world from the ashes. The colors blend, swirl, and merge, creating an immersive experience that defies the boundaries between reality and fantasy. "Phoenix Rising" stands as a testament to the transformative power of street art, a beacon of hope in a city that never sleeps, a mural that whispers to all who pass by: "Rise from the ashes, for in the darkness lies the seeds of rebirth." . street art, vibrant, urban, detailed, tag, mural" ,

**High-Fidelity Photorealistic Subsets (Flux_photo, SD3_photo):**

- *Target:* High-quality photorealism, including human portraits.
- *Prompting:* Prompts were generated by Llama-3.1 based on a structured template inspired by community best practices for detailed photo generation. The template format was: '[STYLE OF PHOTO] photo of a [SUBJECT], [IMPORTANT FEATURE], [MORE DETAILS], [POSE OR ACTION], [FRAMING], [SETTING/BACKGROUND], [LIGHTING], [CAMERA ANGLE], [CAMERA PROPERTIES], in style of [PHOTOGRAPHER]'. Llama-3.1 filled these placeholders randomly using the system instruction: "You are a prompt generator for text to image synthesis, now you should randomly generate a prompt for me to generate an image. The prompt requirements is: [prompt_format]. Prompt only, no more additional reply".
- *Scale:* As per Table 7.
- *Example:* "Moody black and white photo of a woman, dressed in a flowing Victorian-era gown, holding a delicate, antique music box, standing on a worn, wooden dock, overlooking a misty, moonlit lake, with a faint, lantern glow in the distance, in the style of Ansel Adams."

  "High-contrast black and white photo of a majestic lion, with a strong mane and piercing eyes, standing proudly on a rocky outcrop, in mid-stride, with the African savannah stretching out behind it, bathed in warm golden light, from a low-angle camera perspective, with a wide-angle lens, in the style of Ansel Adams."

## C.4 Rationale as a Test-Only Benchmark

GenImage++ is intentionally a **test-only** dataset and does not include corresponding real images for each generated category. This design choice is due to several factors:

1. **Copyright and Sourcing Challenges:** Sourcing perfectly corresponding, high-quality real images for the vast array of styles, complex scenes, and specific portrait attributes generated by modern models poses significant copyright and practical challenges, especially avoiding the use of web-scraped data that may itself contain AI-generated content or copyrighted material. Using reverse image search is often infeasible for highly stylized or uniquely composed generations.

2. **Focus on Generalizable Artifacts:** Our primary aim is to test a detector's ability to identify intrinsic generative artifacts ($f_{gen}$) that are independent of specific semantic content.

   - For the *Base Label* and *Realistic Long-Prompt* subsets, the underlying semantic categories are largely derived from ImageNet, for which abundant real image counterparts already exist within the original GenImage dataset and other standard vision datasets.
   - For the *Multi-Style* subsets, our experiments (Section 6.4) show that detectors trained on SDv1.4 can easily generalize to SDv1.5-generated styles but struggle immensely with styles from Flux or SDXL. This suggests the difficulty lies not in recognizing the styled semantic content (which would be a $c$-dependent feature) but in identifying the differing generative artifacts of the underlying models.

Therefore, GenImage++ serves as a robust benchmark for evaluating a detector's generalization to new generator architectures and advanced prompting techniques, focusing on the detection of AI-specific fingerprints rather than simple real-vs-fake comparison on matched content. **Beyond the generated images themselves, all code used for data generation, along with the structured JSON files containing the prompts for each subset, will be made publicly available to facilitate further research and benchmark replication.**

## D Extended Discussion and Future Directions

### D.1 Broader Relevance of Robustness to Latent Perturbations

The robustness against perturbations in the initial latent space $z_T$ holds significance beyond detecting standard text-to-image outputs generated from purely random noise. Several increasingly common AIGC tasks involve non-standard or optimized initial latent variables:

- **Image Editing and Inpainting:** These tasks often initialize the diffusion process from a latent code obtained by inverting a real image, potentially modifying this latent before generation [28].
- **Optimized Initial Noise:** Recent work focuses on finding "better" initial noise vectors $z_T$ than random Gaussian samples to improve generation quality or efficiency, sometimes referred to as "golden noise" [47].

A detector solely robust to outputs from standard Gaussian noise $z_T^{\mathrm{rand}}$ might be vulnerable when encountering images produced in these scenarios. Therefore, by adversarially training our detector using examples derived from optimized latents $z_T^{\mathrm{adv}}$, we not only aim to improve generalization by forcing the learning of $f_{gen}$, but also potentially enhance the detector's applicability and robustness across this wider spectrum of AIGC tasks that utilize non-standard latent initializations. This presents an interesting avenue for future investigation.

### D.2 Limitations

While our proposed on-manifold latent optimization attack and subsequent adversarial training demonstrate significant improvements in AIGC detector generalization, we acknowledge certain limitations in the current study.

**Computational Cost of Latent Optimization Attack.** Our white-box attack methodology, which optimizes the initial latent noise $z_T$, requires maintaining the computational graph through each denoising step of the diffusion model to backpropagate gradients from the detector's loss back to $z_T$. This incurs a substantial VRAM cost. For instance, attacking a detector using Stable Diffusion v1.4 with 35 denoising steps consumed approximately 79GB of VRAM on an NVIDIA A100 GPU. This high memory requirement currently limits the scalability of generating very large adversarial datasets using this specific attack technique or applying it to even larger generator models without access to high-end hardware.

**Requirement for Differentiable Preprocessing and Detector Retraining.** To ensure end-to-end differentiability from the detector's loss back to the initial latent $z_T$, all intermediate image preprocessing steps (e.g., resizing, normalization) must also be differentiable. In our implementation, this necessitated the use of libraries like Kornia [9] for these transformations, rather than potentially

more common, non-differentiable pipelines involving libraries like Pillow or standard 'torchvision.transforms' if they break the gradient chain during the attack generation. Consequently, when generating adversarial examples against a specific target detector, or when performing adversarial training, the detector ideally needs to be trained or fine-tuned within a framework that utilizes such differentiable preprocessing (as our baseline detectors were). This can limit the direct "out-of-the-box" application of our attack generation process to arbitrary pre-trained third-party detectors if their original training and inference preprocessing pipelines are not fully differentiable or easily replicable with differentiable components. While we retrained baselines like ResNet50 and CLIP+Linear within this Kornia-based framework for fair evaluation and adversarial training, evaluating a wide array of externally pre-trained detectors (whose exact preprocessing might be unknown or non-differentiable) with our specific latent attack would require re-implementing or adapting their input pipelines.

Future work could explore methods to reduce the VRAM footprint of the latent attack, perhaps through gradient checkpointing within the diffusion model during attack or by developing more memory-efficient approximation techniques. Additionally, investigating strategies to adapt our latent attack or on-manifold adversarial examples to detectors with non-differentiable preprocessing pipelines could further broaden the applicability of our findings.

### D.3 Broader Impacts

The research presented in this paper aims to advance the robustness and generalization of AIGC detection, which is crucial for mitigating potential misuse of AI-generated content, such as disinformation or non-consensual imagery. By identifying and addressing vulnerabilities like latent prior bias, we hope to contribute to more reliable forensic tools.

However, as with any research involving adversarial attacks, there is a potential for the attack methodology itself to be misused. The on-manifold latent optimization attack described could, in principle, be employed by malicious actors to craft AIGC that evades detection systems. We acknowledge this possibility but believe the direct negative impact of our specific attack method is likely limited due to several factors:

- **White-Box Requirement:** Our attack is a white-box method, requiring full access to the target detector, including its architecture, weights, and the ability to compute gradients with respect to its inputs. This significantly restricts its applicability against closed-source or API-based detection services where such access is not available.
- **Computational Cost:** As discussed in Section D.2, generating adversarial examples via latent optimization is computationally intensive, particularly in terms of VRAM. This may act as a practical barrier for large-scale misuse by those without significant computational resources.

Furthermore, the primary contribution of this work lies in using these attacks as a diagnostic tool to understand detector weaknesses (latent prior bias) and subsequently as a means to *improve* detector robustness through on-manifold adversarial training (OMAT). The insights gained are intended to strengthen defenses, and the OMAT paradigm itself offers a pathway to more resilient detectors. We will also make our GenImage++ benchmark publicly available, which will aid the community in developing and evaluating more robust detectors against contemporary generative models.

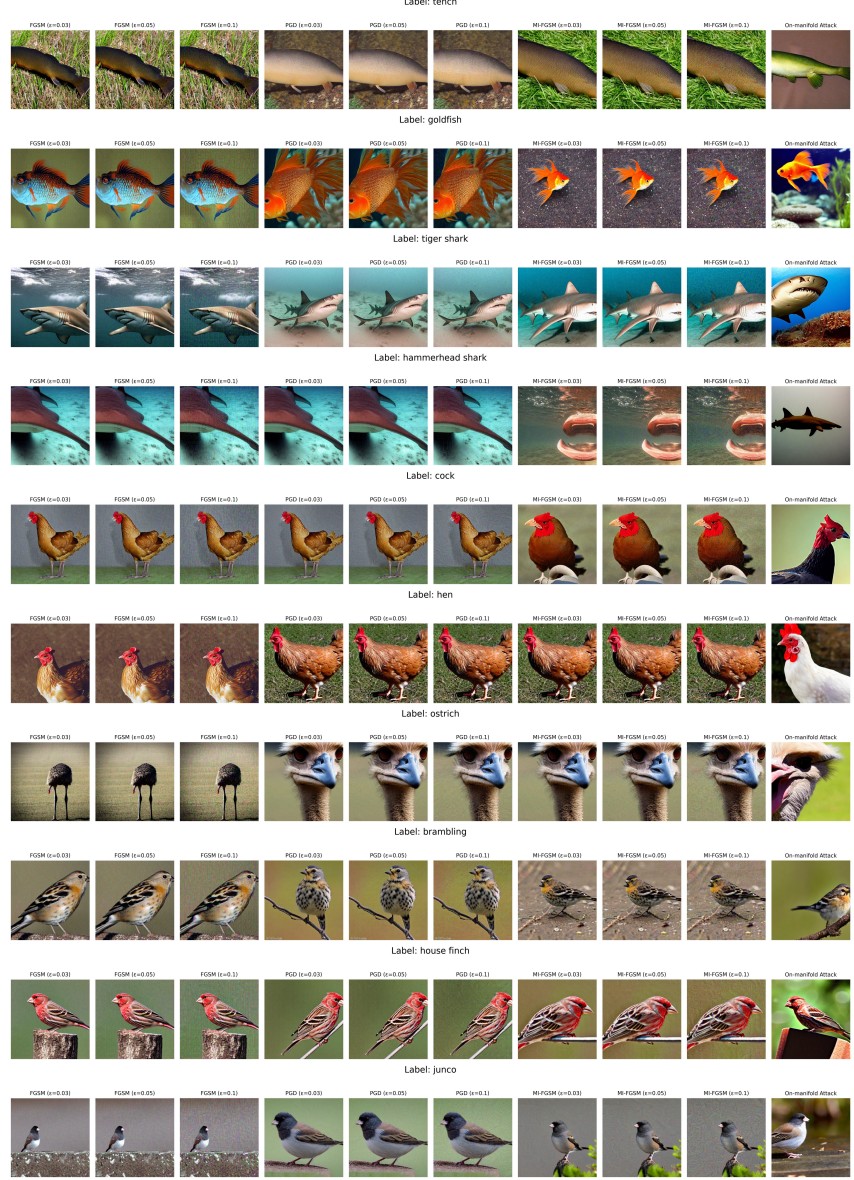

Figure 8: Qualitative comparison of adversarial examples generated for various ImageNet labels. Each row corresponds to a different label. Columns display results from different $L_\infty$-bounded pixel-space attacks (FGSM, PGD, MI-FGSM with varying $\epsilon$) and our proposed on-manifold latent optimization attack (rightmost column). Pixel-space attacks often introduce visible noise, especially at higher $\epsilon$, while our on-manifold attack maintains high visual fidelity consistent with the generator's capabilities.

