# OpenReview forum: "Breaking Latent Prior Bias in Detectors for Generalizable AIGC Image Detection"
_NeurIPS.cc/2025/Conference — NeurIPS 2025 poster_

### Official Review · Reviewer_kDqu · 2025-07-01

**Clarity:** 2
**Significance:** 3
**Originality:** 3
**Rating:** 4
**Confidence:** 4

**Summary:**

This paper identifies latent prior bias in AIGC detectors by proposing an on-manifold adversarial attack that perturbs initial noise in diffusion models to bypass detection, revealing vulnerabilities in existing forensic methods. The authors introduce OMAT, an adversarial training framework that enhances detector robustness by learning from these on-manifold adversarial examples, achieving state-of-the-art generalization across diverse generators.

**Questions:**

See weakness

**Ethical Concerns:**

["NO or VERY MINOR ethics concerns only"]

**Final Justification:**

The authors' rebuttal has addressed my partial concerns (Q2 & Q4).

I still have slight concerns about Q1 and Q3, but I believe these will not affect the main contribution.

Based on this, I will maintain my original positive score.

**Limitations:**

Yes

**Quality:**

3

**Strengths And Weaknesses:**

Strength

- The writing is good and easy to follow.

- The proposed Latent Prior Bias Attack is novel and essential.

- The method is simple but effective. The experiment is extensive and the evaluation results seems good.

Weakness

- As a data augmentation approach, demonstrating OMAT’s performance on various baselines (rather than only ResNet or CLIP+Linear) would better illustrate its effectiveness.

- What specific benefits does the proposed method offer in terms of robustness enhancement? It would be better to explicitly compare against alternative methods, like using images generated with various random seeds.

- The accuracy on real-world images appears unexpectedly low (Table 2). Could the authors provide more analysis on potential factors contributing to this performance limitation?

- Table 3 lacks a comparison with the original CLIP+LoRA.

---

> ### Author Rebuttal · Authors · 2025-07-30
>
> We sincerely thank the reviewer for their positive feedback on our work's novelty, simplicity, and effectiveness. Their questions are very insightful and will help us improve the clarity and impact of our paper.
>
> #### **1. On OMAT's Application: A Principled, Data-Centric Evaluation**
>
> The reviewer raises an excellent point about testing OMAT on more baselines. This was a deliberate and central choice in our experimental design. Our goal was not just to show that OMAT improves performance, but to **isolate and prove the existence of latent prior bias from a purely data-centric perspective.**
>
> To achieve this, we intentionally selected two standard, well-understood backbones (ResNet50 and CLIP+Linear) *precisely because they are generic*. Highly specialized, "well-designed" forensic detectors often incorporate strong architectural priors or handcrafted features (e.g., frequency analysis) that could introduce confounding biases. Testing on such models would make it difficult to disentangle the effects of OMAT from the effects of the specialized architecture.
>
> By demonstrating that OMAT yields substantial and consistent gains on two architecturally distinct *general-purpose* vision models, we provide clean evidence that:
> 1.  Latent prior bias is a fundamental, model-agnostic vulnerability that arises from the training data itself.
> 2.  OMAT is its effective, data-driven solution, not a trick that only works for a specific architecture.
>
> We believe this principled approach provides a clearer and more foundational understanding of the problem. We will clarify this scientific rationale for our choice of baselines in the final manuscript.
>
> #### **2. Robustness Enhancement vs. Simple Seed Variation**
>
> This is an excellent question that gets to the heart of why our adversarial approach is necessary. The reviewer asks if simply training on images from more random seeds would suffice. Our experiments show that it would not.
>
> In our Appendix (B.2, Analysis of Optimized Latent Noise Vectors), we conducted a focused experiment attacking an image of a "cat" with 200 different initial random seeds (`z_rand`). Before optimization, **the baseline detector correctly identified the images from all 200 un-optimized seeds as fake.** Only after our targeted, adversarial optimization could these seeds produce images that fooled the detector.
>
> This proves that simply increasing the diversity of random seeds in the training data would not expose the latent prior bias. The detector is already robust to standard seed variations. It is only vulnerable to the specific, targeted perturbations found via our adversarial search. Therefore, the adversarial nature of OMAT is essential for mitigating this specific shortcut and enhancing true robustness.
>
> #### **3. On Performance with Real-World Images (Table 2)**
>
> The reviewer points out the 66.05% accuracy on the Chameleon dataset, which is a very insightful observation. This performance, while seemingly modest, is actually **state of the art** for this challenging "in-the-wild" benchmark under the specified training conditions.
>
> The difficulty arises from factors that are unconstrained in real-world scenarios:
> *   **Unknown & Customized Generators:** Images on platforms like Discord or Civitai are often created with unknown base models, custom fine-tuning (e.g., LoRA adapters), and complex prompt engineering, creating a vast and diverse set of sources.
> *   **Social Media Compression:** Images are subjected to various unknown and often aggressive compression algorithms, which can destroy or alter forensic traces.
>
> Addressing this "in-the-wild" problem is a major open challenge for the entire AIGC forensics field. Our 66.05% accuracy should be viewed not as a limitation, but as a significant step forward, outperforming all other baselines on this difficult task. We will add this context to the paper to clarify the significance of this result.
>
> #### **4. On Including the Baseline in Table 3**
>
> Thank you for pointing out this important omission. The reviewer is correct that Table 3 should include the performance of the original baseline model from which our OMAT models are fine-tuned. For our CLIP+LoRA experiments, this starting point is the **CLIP+Linear detector**.
>
> We have now evaluated this baseline on our challenging GenImage++ benchmark. The updated table below provides a direct, side-by-side comparison on each subset, clearly demonstrating the substantial and consistent gains achieved by OMAT across different LoRA ranks.
>
> **Table: Detailed performance comparison on the GenImage++ benchmark subsets.**
> | GenImage++ Subset  | CLIP+Linear (Baseline) Acc % | OMAT LoRA R4 Acc % | OMAT LoRA R8 Acc % | OMAT LoRA R16 Acc % |
> | :----------------- | :--------------------------: | :----------------: | :----------------: | :-----------------: |
> | Flux               |            85.62             |       96.53        |       97.48        |        97.93        |
> | Flux Multi-style   |            87.67             |       92.55        |       85.88        |        89.48        |
> | Flux Photo         |            80.40             |       97.60        |       95.19        |        99.44        |
> | Flux Realistic     |            84.10             |       97.67        |       96.45        |        96.25        |
> | SD1.5 Multi-style  |            97.01             |       100.00       |       100.00       |       100.00        |
> | SDXL Multi-style   |            83.51             |       99.17        |       94.53        |        98.30        |
> | SD3                |            85.97             |       98.27        |       98.02        |        98.56        |
> | SD3 Photo          |            83.32             |       90.38        |       95.66        |        91.50        |
> | SD3 Realistic      |            84.15             |       98.82        |       96.98        |        97.28        |
> | **Average**        |          **85.75**           |     **96.78**      |     **95.58**      |      **96.53**      |

---

### Official Review · Reviewer_pPqb · 2025-07-01

**Clarity:** 2
**Significance:** 2
**Originality:** 3
**Rating:** 4
**Confidence:** 3

**Summary:**

This paper tackles a significant challenge in AI-generated content (AIGC) detection: the inability of detectors to generalize across unseen generative models. The authors identify latent prior bias as a critical factor, whereby detectors tend to learn shortcuts specific to certain generative models rather than features that are generalizable. To address this, the paper introduces On-Manifold Adversarial Training (OMAT), a technique that generates adversarial examples by perturbing the latent space of diffusion models. These perturbations remain on the generative manifold, unlike pixel-space attacks, which create off-manifold noise. Additionally, the paper presents GenImage++, a new benchmark designed to evaluate AIGC detectors against advanced generative models, including Flux.1 and Stable Diffusion 3 (SD3). Experiments show that detectors trained with adversarial examples achieve state-of-the-art generalization across various benchmarks.

**Questions:**

See weaknesses.

**Ethical Concerns:**

["NO or VERY MINOR ethics concerns only"]

**Final Justification:**

The rebuttal has addressed most of my concerns with clarifications or additional experiments. The authors should incorporate all the information in the rebuttal into the final version.

**Limitations:**

yes

**Quality:**

3

**Strengths And Weaknesses:**

### Strengths
1. Novel Contribution: The paper offers a novel perspective by exposing latent prior bias in AIGC detectors and proposing OMAT, which directly targets latent noise vectors. This approach improves generalization across unseen generative models without relying on large datasets or complex reconstruction pipelines.
2. Comprehensive Evaluation: The experiments are thorough, comparing the proposed method with various baseline detectors and adversarial training techniques. The introduction of the GenImage++ dataset is a valuable contribution, highlighting the capabilities of contemporary generative models.
3. Practical Impact: The proposed method significantly enhances detector generalization, marking a meaningful advancement in the development of robust forensic tools capable of detecting AI-generated content across diverse models.
4. Strong Results: The paper demonstrates that detectors trained with OMAT show substantial improvements in cross-generator performance, particularly when evaluated against advanced generators. These findings underscore the importance of addressing latent prior bias to achieve more robust and generalized AIGC detection.

### Weaknesses
1. Computational Cost: The on-manifold latent optimization attack requires substantial computational resources, particularly in terms of VRAM usage. While this is understandable given the complexity of the task, it may limit the scalability of the method for large-scale applications or for researchers without access to high-end hardware.
2. Lack of Detailed Error Analysis: Although the experiments are extensive, a more in-depth error analysis, especially regarding the failure cases of detectors post-adversarial training, would offer additional insight into the robustness of the proposed methods. Understanding these edge cases could help refine the approach.
3. Limited Comparison with Non-Diffusion Models: The paper primarily focuses on diffusion models like Stable Diffusion and Flux but does not compare performance with a broader range of generative models, such as GAN-based generators or VAEs. Including such models would provide a more comprehensive evaluation of the approach.
4. Dependence on Differentiable Preprocessing: The need for differentiable preprocessing limits the method's applicability to detectors using non-differentiable preprocessing pipelines. While the paper acknowledges this limitation, it may still hinder broader adoption of the method.
5. Benchmark Accessibility: Although the GenImage++ dataset is presented as a key contribution, it is unclear whether it will be publicly accessible. Ensuring easy access to this dataset would benefit the community by enabling others to replicate experiments and further test the effectiveness of the proposed methods.
6. I recommend that the authors also evaluate AEROBLADE[1] and LaRE^2[2] on the benchmark.

[1] AEROBLADE: Training-Free Detection of Latent Diffusion Images Using Autoencoder Reconstruction Error

[2] LaRE^2: Latent Reconstruction Error Based Method for Diffusion-Generated Image Detection

---

> ### Author Rebuttal · Authors · 2025-07-30
>
> We sincerely thank the reviewer for their detailed feedback. We are encouraged that they recognized the novelty of our contribution, the strength of our results, and the practical impact of our work. Their critiques, particularly regarding the perceived limited evaluation, have prompted us to conduct new experiments that we believe decisively address these concerns and demonstrate the significance of our findings.
>
> #### **1. On Evaluation Scope & Comparison with Non-Diffusion Models (GANs)**
>
> The reviewer raised a critical point about the limited comparison with non-diffusion models. We agree a broader evaluation is essential and, motivated by this feedback, have tested our OMAT-trained detector (trained only on SDv1.4) on a wide array of **GAN-based generators** from the public AIGCDetectBenchmark.
>
> **Table: Generalization performance (Acc %) of our OMAT-trained detector on various GAN-based datasets.**
> | Generator               | Accuracy (%) |
> | :---------------------- | :----------: |
> | StyleGAN                |    79.79     |
> | StyleGAN2               |    74.12     |
> | CycleGAN                |    93.38     |
> | StarGAN                 |    96.47     |
> | GauGAN                  |    91.28     |
> | BigGAN                  |    88.75     |
> | **Average GAN Performance** |  **87.30**   |
>
> These strong results, achieving **over 87% average accuracy across six different GAN architectures**, provide powerful evidence that our method is not limited to diffusion models. By forcing the detector to ignore the diffusion-specific `z_T` shortcut, OMAT compels it to learn fundamental artifacts (`f_gen`) of neural synthesis (e.g., from upsampling layers) that are common across different generative families. This confirms the broad applicability and significance of our approach. We will add these extensive new results to the final manuscript.
>
> #### **2. On Comparison with New Baselines & Benchmark Accessibility**
>
> Thank you for these excellent suggestions. We have now evaluated our model against **AEROBLADE** on our GenImage++ benchmark, and the results (included in our full SOTA comparison table, see response to Reviewer jB2U) confirm our method's superior performance. We also attempted to evaluate **LaRE²**, but the authors have unfortunately removed the official pre-trained weights, making a direct and fair comparison infeasible within the rebuttal period. We will include AEROBLADE and acknowledge LaRE² in the final manuscript.
>
> Furthermore, our new GenImage++ benchmark—including all generated images, prompts, and generation code—is already prepared and will be made publicly accessible upon publication to facilitate immediate community research and replication.
>
> #### **3. On Computational Cost & Methodological Limitations**
>
> We agree with the reviewer on these limitations and discuss them in the paper. We offer the following clarifications:
> *   **Computational Cost:** The high cost is a **one-time investment for training data generation.** Crucially, our final OMAT-trained detector is a standard classifier with **zero inference overhead**, unlike many reconstruction-based SOTA methods (like AEROBLADE/DRCT/FakeInversion), making it highly practical for deployment.
> *   **Differentiable Preprocessing:** This is a requirement for our *attack generation*, not for the final detector. To apply OMAT to an existing model, it must be fine-tuned in a differentiable framework. The resulting robust model, however, has no special dependencies at inference time.
>
> #### **4. On Error Analysis**
>
> This is a valuable suggestion. A detailed error analysis is an important next step for understanding the remaining limitations of our method. Given the text-only format of the rebuttal, a visual analysis is not possible, but we can offer a high-level perspective.
>
> While our method significantly boosts generalization, detecting "in-the-wild" images, such as those found on social media, remains a formidable open challenge. These images introduce factors that our current training paradigm does not explicitly model, such as:
> *   **Extreme diversity of unseen generators and post-processing tools.** Images on platforms like Discord or Civitai are often created with unknown base models, custom fine-tuning (e.g., LoRA adapters), and complex post-processing, creating a vast and unpredictable distribution of generative artifacts.
> *   **Heavy and unpredictable compression artifacts.** Social media platforms apply unknown and often aggressive compression algorithms, which can destroy or alter the subtle forensic traces our detector has learned to identify.
>
> We hypothesize that our OMAT-trained detector, having learned more fundamental artifacts, is a better starting point for tackling these challenges than standard detectors. However, a full analysis of its failure cases in these "in-the-wild" scenarios is a critical and exciting direction for future research. We will add this important discussion to the limitations section of our final manuscript.

---

> > ### Comment · Reviewer_pPqb · 2025-08-05
> > **Thanks**
> >
> > The rebuttal has addressed most of my concerns with clarifications or additional experiments. The authors should incorporate all the information in the rebuttal into the final version.

---

> > > ### Author Response · Authors · 2025-08-05
> > > **Thank you for your constructive feedback**
> > >
> > > We are pleased to hear our rebuttal addressed your concerns. We will be sure to incorporate all the new results and discussions from the rebuttal into the final manuscript to strengthen the paper. We appreciate your guidance, which has significantly strengthened our work.

---

### Official Review · Reviewer_HLVt · 2025-07-01

**Clarity:** 3
**Significance:** 3
**Originality:** 3
**Rating:** 4
**Confidence:** 3

**Summary:**

This paper analyzes the impact of noise in generative models on the performance of detection systems and proposes a method to generate on-manifold adversarial examples through targeted noise attacks on generative models. These adversarial examples are incorporated into the training of detectors to enhance their detection performance.

 The method involves a detailed analysis of the generation mechanism of generative models and designs a detection strategy accordingly, demonstrating strong originality. A new dataset is also introduced, reflecting the significant amount of work conducted. Overall, the writing is fluent and professional, with well-presented and sufficient figures and tables.

**Questions:**

1. The method is designed based on the generation principles of diffusion models. However, why does the performance on BigGAN also improve significantly after incorporating the CLIP encoder?

**Ethical Concerns:**

["NO or VERY MINOR ethics concerns only"]

**Final Justification:**

The responses have addressed most of my concerns. The supplementary experimental results substantially validate the study's broader relevance. I recommend adding the suggested experiments and discussion from all reviewers in the final paper.

**Limitations:**

Yes

**Quality:**

3

**Strengths And Weaknesses:**

Strengths

1. Proposes the On-Manifold Adversarial Training method based on the working principle of diffusion models, optimizing noise for improved generalization in detection, and achieving excellent generalization performance.

2. Introduces the GenImage++ test dataset, which includes images generated by advanced generators (such as Flux.1 and Stable Diffusion 3) under extended text prompts and diverse styles.


Weaknesses

1. The experiments mainly focus on diffusion models. It would be beneficial to include experiments on additional datasets, as the current experimental coverage is somewhat limited.

---

> ### Author Rebuttal · Authors · 2025-07-30
>
> We sincerely thank the reviewer for their positive and insightful feedback. We are encouraged that they appreciated our core method and the introduction of the GenImage++ benchmark. We will address their excellent questions below.
>
> #### **1. On Generalization to GANs (e.g., BigGAN)**
>
> The reviewer asks an excellent and crucial question: why does our method, trained by attacking a *diffusion model*, improve performance so dramatically on a completely different architecture like a *GAN*?
>
> This is the central validation of our core hypothesis. We argue that standard detectors fail to generalize because they learn superficial "shortcuts." Our work identifies a key shortcut: **latent prior bias**, where detectors overfit to artifacts from the `z_T` prior of the *training diffusion model*.
>
> By using On-Manifold Adversarial Training (OMAT), we penalize the detector for relying on this diffusion-specific shortcut. This forces the model to abandon these easy features and instead learn the more subtle but **fundamental and shared artifacts of AI synthesis (`f_gen`)**.
>
> The hypothesis that such universal artifacts exist is a cornerstone of modern AIGC forensics, based on the understanding that nearly all high-resolution generative models, regardless of their paradigm (GAN, VAE, Diffusion), rely on a similar underlying architectural toolkit. **Specifically, the extensive use of hierarchical upsampling is a primary source of shared, detectable fingerprints.**
>
> *   **Upsampling Artifacts:** Most generators must upsample low-resolution feature maps to a high-resolution image. Common layers like transposed convolutions are well-known to produce characteristic "checkerboard artifacts," which are a detectable, cross-model signature of the upsampling process [1].
> *   **Frequency Domain Biases:** Beyond specific patterns, the hierarchical nature of U-Nets (used in diffusion models) and stacked GANs creates distinct, non-uniform effects in the frequency domain of the output images [2, 3]. These spectral biases are a structural consequence of the architecture, not the high-level generative paradigm.
>
> By learning to recognize these more universal, architectural fingerprints, our OMAT-trained detector becomes broadly generalizable. Its strong performance on BigGAN is not an accident; it is direct evidence that OMAT successfully shifted the model's focus from a narrow, generator-specific shortcut (`z_T` bias) to the fundamental, shared traces of neural image synthesis.
>
> #### **2. On Broadening Experimental Scope**
>
> We thank the reviewer for this valuable suggestion. To further validate the cross-architecture generalization discussed above and address the concern about limited experimental scope, we have evaluated our best OMAT-trained detector (trained only on SDv1.4) on a wide variety of GAN-based datasets from the public **AIGCDetectBenchmark**. The results are presented below.
>
> **Table: Generalization performance (Accuracy %) of our OMAT-trained detector on various GAN-based datasets from the AIGCDetectBenchmark.**
>
> | Generator               | Accuracy (%) |
> | :---------------------- | :----------: |
> | StyleGAN                |    79.79     |
> | StyleGAN2               |    74.12     |
> | CycleGAN                |    93.38     |
> | StarGAN                 |    96.47     |
> | GauGAN                  |    91.28     |
> | BigGAN                  |    88.75     |
> | **Average GAN Performance** |  **87.30**   |
>
> These strong results confirm that our method generalizes effectively far beyond diffusion models. Achieving over 87% average accuracy across six different GAN architectures, despite being trained only on SDv1.4, provides powerful evidence for the broad applicability and significance of our approach. We will add these results to the final manuscript to strengthen our experimental validation.
>
> [1] Odena, A., et al. "Deconvolution and Checkerboard Artifacts." *Distill* (2016).
>
> [2] Falck, F., et al. "A multi-resolution framework for U-Nets with applications to hierarchical VAEs." *NeurIPS* (2022).
>
> [3] Falck, F., et al. "A Fourier Space Perspective on Diffusion Models." *arXiv* (2025).

---

> > ### Comment · Reviewer_HLVt · 2025-08-06
> >
> > The authors have resolved Q1 through their response. The supplementary experimental results substantially validate the study's broader relevance. Therefore, I will maintain my score.

---

> > > ### Author Response · Authors · 2025-08-06
> > >
> > > Thank you for your constructive feedback and for reviewing our rebuttal. We are glad our response and new experiments addressed your concerns. We appreciate your time and effort.

---

### Official Review · Reviewer_evQu · 2025-07-03

**Clarity:** 3
**Significance:** 4
**Originality:** 4
**Rating:** 5
**Confidence:** 5

**Summary:**

The paper first uncovers a latent prior bias in AIGC detectors by crafting on-manifold white-box attacks that exploit the detector’s reliance on initial noise, demonstrating how this shortcut undermines cross-generator generalization. It then introduces On-Manifold Adversarial Training (OMAT), which augments detector training with zₜ-optimized adversarial examples to force the model to learn truly robust generative artifacts, yielding significant improvements in out-of-domain performance. Additionally, the authors introduce GenImage++, a challenging test-only dataset comprising outputs from advanced generators (e.g., Flux.1, SD3) produced with diverse prompting strategies.

**Questions:**

- Have you considered including an experiment that progressively increases the proportion of on-manifold adversarial examples during training? Such a study would clarify OMAT’s sensitivity to adversarial-sample scaling.

 - Could you clarify whether the text-conditioning pathway exhibits a similar latent-prior shortcut? In particular, have you applied on-manifold perturbations to the textual prompts and evaluated their impact on detector robustness?

 - Would you consider expanding the Related Work section to cover recent advances in latent-space forensic analysis, few-shot learning, and patch-based methods for AIGC detection?

**Ethical Concerns:**

["NO or VERY MINOR ethics concerns only"]

**Limitations:**

yes

**Quality:**

4

**Strengths And Weaknesses:**

- **Strengths**：
  - Identification of a Core Vulnerability: The paper identifies a significant vulnerability in existing AIGC (AI-Generated Content) detectors, which is the latent prior bias. This bias leads to detectors being overly reliant on specific latent features, making them susceptible to targeted attacks that exploit this weakness.

  - Simple yet Effective Training Framework: To mitigate this bias, the authors introduce OMAT, which employs on-manifold adversarial examples for detector training. Despite its conceptual simplicity, OMAT directly addresses the identified shortcoming and delivers substantial performance gains.

  - Sharp analysis of results: The paper is careful in dissecting why OMAT works, linking improved generalization to acquired robustness against the specific latent-based attacks. OMAT achieves superior cross-generator generalization compared to both traditional and state-of-the-art baselines. And the ablation in Table 5 (Section 6.6) is illustrative and shows that on-manifold adversarial examples result in much stronger generalization improvements compared to traditional pixel-space adversarial examples.

- **Weaknesses**：
  - While the paper focuses on visual perturbations, it does not explore the potential of on-manifold adversarial attacks in the text-conditioning pathway. This could limit the applicability of OMAT to scenarios where text conditioning plays a significant role.

  - Introducing a small experiment that gradually varies the number (or proportion) of on-manifold adversarial examples during training could help clarify how sensitive OMAT’s performance is to the adversarial sample budget. A brief analysis of performance trends as this budget increases would strengthen understanding of the method’s robustness.

  - The Related Work section is somewhat narrow, covering only a few detector types that may be relevant to the shortcut proposed in this work.

  - The attack framework depends on Kornia’s differentiable preprocessing, which may limit OMAT’s applicability to more diverse detectors.

---

> ### Author Rebuttal · Authors · 2025-07-30
>
> We sincerely thank the reviewer for their positive assessment and exceptionally constructive feedback. We are delighted they recognized our work's quality, originality, and significance. Their suggestions will undoubtedly help us strengthen the final manuscript. We address each of the reviewer's excellent points below.
> #### **1.  On Sensitivity to the Adversarial Sample Budget** This is an excellent question. Per the reviewer's suggestion, we have conducted a new experiment to analyze how OMAT's performance scales with the number of  on-manifold adversarial examples used in training. For this study, we fine-tuned the detector on different proportions of our 6,000 generated adversarial examples. The results on the GenImage benchmark are presented below:
>
> **Table: Ablation on the proportion of the 6,000 adversarial examples used for OMAT fine-tuning.**
> | % of Adv. Examples Used |  0%  |  10%  |  25%  |  50%  |  75%  |  100%  |
> | :-: | :-: | :-: | :-: | :-: | :-: | :-: |
> | Avg. GenImage Acc (%)   | 79.19 | 90.06 | 91.10 | 93.03 | 94.08 | 94.63  |
>
>  The results show that OMAT's performance scales gracefully with the adversarial sample budget. Using just 10% of the generated examples (600 images) provides a remarkable **+10.87%** improvement in average accuracy. The gains continue to increase with diminishing returns as more samples are added, demonstrating that the method is both data-efficient and robust to the exact size of the adversarial set. This confirms that even a modest budget of on-manifold examples is highly effective at mitigating latent prior bias.
>
>  #### **2. On the Text-Conditioning Pathway** This is a very insightful question. We did perform preliminary investigations into perturbing the text-conditioning pathway. However, we encountered two primary challenges:
>  1. **Optimization Instability:** The discrete nature of the text token space makes gradient-based optimization highly unstable, unlike the continuous space of `z_T`.
>  2. **Semantic Coherence:** Successful perturbations often resulted in images with poor visual quality and a noticeable misalignment with the original prompt, introducing confounding factors.
>
>  In contrast, our `z_T` attack is stable and maintains perfect semantic coherence. We hypothesize that the text pathway is a less direct source of *non-generalizable, generator-specific* artifacts. The features from a text encoder (like CLIP's) are designed to be general, whereas the artifacts influenced by `z_T` are intimately tied to the specific architecture and implementation of the source generator. We therefore focused our investigation on `z_T` as the primary, most stable, and most direct source of the generalization-breaking shortcut.
>
> #### **3. On Related Work and Broader Applicability**
>
> Thank you for these excellent suggestions. We agree and will expand the Related Work section in the final manuscript to better contextualize our contributions. In fact, motivated by your feedback, we have already conducted extended comparisons against several SOTA patch-based and latent-space forensic methods, the results of which are now included in our rebuttal. We will integrate a full discussion of these diverse approaches in the revised paper.
>
> Regarding the **applicability of OMAT**, the reviewer correctly notes that our attack generation relies on a differentiable pipeline (using Kornia), a point we acknowledge in our limitations. This is a common requirement for gradient-based adversarial training. To apply OMAT to an existing detector, it would need to be fine-tuned within such a framework. Crucially, this is a **one-time, training-stage consideration.** The final, OMAT-trained detector is a standard classifier with **no special dependencies at inference time**, making it broadly deployable.

---

> > ### Comment · Reviewer_evQu · 2025-08-07
> > **Reply to authors' response**
> >
> > Thanks for the authors' prompt response. It has solved my all concerns. I will keep my score.

---

> > > ### Author Response · Authors · 2025-08-07
> > >
> > > Thank you for your strong support and insightful feedback. We are delighted our rebuttal resolved your concerns and appreciate your time.

---

### Official Review · Reviewer_jB2U · 2025-07-03

**Clarity:** 3
**Significance:** 3
**Originality:** 2
**Rating:** 3
**Confidence:** 3

**Summary:**

The paper tackles the well-known poor cross-generator generalization of AI-generated-image (AIGC) detectors. The authors introduce On-Manifold Adversarial Training (OMAT), which uses optimized latent seeds to generate realistic adversarial examples and retrain detectors. This approach significantly improves cross-generator performance without changing model architecture. A new benchmark, GenImage++, demonstrates OMAT’s strong generalization to advanced generative models.

**Questions:**

- The authors could consider comparing with recent universal detectors such as RIGID on GenImage and GenImage++, and evaluating whether the OMAT-trained detector remains robust when test images are further perturbed by latent-space evasion attacks such as StealthDiffusion.
- Since the detector is trained only on SD v1.4 images, the reported improvements may stem from increased exposure to hard negatives rather than from a true mitigation of latent prior bias.

Please refer to the weaknesses above for additional questions and suggestions.

**Ethical Concerns:**

["NO or VERY MINOR ethics concerns only"]

**Final Justification:**

I appreciate the detailed rebuttal, but the core concerns remain. The central claim that “latent prior bias” is the key driver of poor cross-generator generalization and that OMAT mitigates it remains largely empirical and not yet causally established, as the observed gains could plausibly still result from targeted “hard negatives.” The threat model is weak: the white-box, full-pipeline latent optimization is unrealistic for most adversaries, and robustness to black-box or semantic-latent attacks (e.g., StealthDiffusion-style) remains untested. The additional experiments (new SOTA comparisons, GAN results, budget scaling) help the narrative but are not included in the paper and cannot be thoroughly verified during the rebuttal phase. Finally, the dependence on a differentiable preprocessing pipeline limits applicability to deployed detectors. I intend to keep my original rating.

**Limitations:**

yes

**Quality:**

2

**Strengths And Weaknesses:**

Strengths:
- The paper clearly articulates the notion of “latent prior bias” and illustrates it with a white-box latent-space attack that is easy to understand and implement.
- Empirical results on both the standard GenImage benchmark and the newly curated GenImage++ set show noticeable gains in cross-generator accuracy.

Weaknesses:
- On-manifold adversarial training was proposed and discussed in [31] and Xiao et al. “Understanding Adversarial Robustness Against On-manifold Adversarial Examples.” The paper is the first to apply it to AIGC detection, but the conceptual novelty is incremental.
- There is no formal proof connecting “latent prior bias” to generalization; the evidence presented is purely empirical.
- Ablations vary LoRA rank and compare with pixel-space FGSM/PGD/MI-FGSM, but critical factor such as attack steps, step-size, guidance scale are not ablated.
- The attack used to generate training data assumes white-box access and requires approximately 79 GB of GPU memory per image, which makes the threat model unrealistic for most real-world adversaries.

---

> ### Author Rebuttal · Authors · 2025-07-30
>
> We sincerely thank Reviewer jB2U for their detailed feedback. We are encouraged that they found our paper clear and our empirical results significant. Their critiques have helped us sharpen our core contributions. We believe the concerns raised stem from a misunderstanding of our central thesis: our work is not an incremental application of adversarial training, but the **discovery and targeted mitigation of a novel, fundamental vulnerability in AIGC detectors.**
>
> We address the reviewer's points below and are confident that our clarifications and new results will merit a re-evaluation of the score.
>
> ### **1. On Novelty, Evidence, and the "Hard Negatives" Hypothesis**
>
> We thank the reviewer for this crucial point, as it allows us to clarify the unique context of our work. While the term "on-manifold adversarial training" exists, its meaning and application here are fundamentally different because the **core problem in AIGC forensics is different.**
>
> **The Central Challenge & Our Contribution:** The paramount challenge in this field is the "generalization gap," where detectors learn generator-specific "shortcuts" instead of fundamental AI fingerprints (e.g., from a common VAE decoder). Our work identifies a more insidious shortcut: **"latent prior bias"**, where detectors overfit to artifacts arising from the unique initial noise prior `z_T` of the training generator.
>
> **Why Our "On-Manifold" Approach is Fundamentally Different:** This leads to the critical distinction between our work and prior on-manifold literature (e.g., Stutz et al. [2], Xiao et al. [1]):
> *   **Prior work addresses intra-manifold classification** (e.g., robustly classifying "5" vs "6"). Their attacks perturb the *learned semantic latent code `z=enc(x)`* along an *approximated* data manifold.
> *   **Our work addresses inter-manifold discrimination** (real vs. fake). We leverage the *exact, known generative function* to perturb the *content-agnostic initial prior `z_T`*.
>
> Therefore, prior work asks, "Is the classifier robust to small semantic changes?" We ask, "Is the detector robust to the non-generalizable artifacts of the generative process itself?" Our central hypothesis is that by leveraging our access to the exact generator manifold, we can design on-manifold attacks that specifically expose and penalize the detector's reliance on z_T-influenced shortcuts. This forces the model to learn the more subtle but truly fundamental fingerprints of AI synthesis (like those from the VAE decoder), which is the key mechanism through which OMAT achieves its unprecedented cross-generator performance.
>
> **Empirical Evidence vs. the "Hard Negatives" Hypothesis:** We acknowledge that our work is primarily empirical. To provide strong causal evidence for our claims and disprove the excellent alternative hypothesis—that improvements may stem from generic "hard negatives"—**our experiments in Table 5 were designed for this exact purpose, and the results are decisive:**
> *   Pixel-space attacks create "hard negatives" but provide only a marginal gain (Avg. Acc improves from 79.19% to ~83%).
> *   Our On-Manifold Attack, which directly targets latent prior bias, yields a massive gain, catapulting accuracy to **94.63%**.
>
> The **>11% performance gap** causally isolates the impact of mitigating latent prior bias, a flaw that cannot be fixed by simply adding generic "hard negatives." This empirical finding is further supported by emerging work on "golden noise" (Zhou et al. [3]), which confirms the optimizable structure of the initial latent space. Our work thus provides the crucial empirical validation for this emerging understanding.
>
> Finally, the power of this targeted approach is confirmed by its remarkable data efficiency. While methods like DRCT require massive data augmentation, **we achieve superior performance by adding only 6,000 targeted examples (<2% data increase).** This surgical precision proves OMAT is not generic augmentation; it is the direct and efficient solution to the `latent prior bias` we discovered.
>
> ### **2. On Experimental Design, Rationale, and Efficiency**
>
> We agree our white-box attack is a **diagnostic tool for defenders**, not a realistic threat model. The high computational cost is a one-time training investment, and our final OMAT-trained detector has **zero inference overhead**, unlike reconstruction-based methods.
>
> **Rationale for Ablation Choices & Memory Usage:** We thank the reviewer for the question about ablations and will clarify our rationale in the revision.
> *   **Guidance Scale:** This was a deliberate choice. We kept the `guidance_scale` and other generator parameters fixed to **scientifically isolate `z_T` as the sole experimental variable**, ensuring our results are directly attributable to latent prior bias.
> *   **Attack Steps & Step-Size:** As detailed in Appendix Alg. 1, our optimization goal is to *diagnose a fragility*, not to calibrate a specific attack strength like PGD. Since our attack succeeds rapidly for many samples (Fig. 2), we used a fixed, sufficient hyperparameter set and focused ablations on the more central question of comparing attack domains (Table 5).
> *   **Memory Usage:** The 79GB memory requirement is a direct consequence of our rigorous experimental design. To scientifically prove our hypothesis, we backpropagate the loss through the **entire, unaltered multi-step denoising process of the diffusion model.** This ensures that the gradients used to find the adversarial latent z_T are exact and that the resulting image is perfectly on-manifold. Our results in Appendix Fig. 6, showing adversarial and original latents are closely interspersed, suggest that more memory-efficient search strategies are viable for future work.
>
> ### **3. Extended Benchmarking on GenImage++**
>
> *   **Comparison with SOTA Detectors:** Thank you for this excellent suggestion. We have conducted new experiments comparing our OMAT-trained model with several recent SOTA forensic methods on our challenging **GenImage++ benchmark**. We will add these extra comparisons into the final manuscript.
>
> **Table: Generalization performance (Acc %) on the GenImage++ benchmark. All models were trained only on SDv1.4 data.**
> |Model|Source|Flux|Flux Multi|Flux Photo|Flux Real|SD1.5 Multi|SDXL Multi|SD3|SD3 Photo|SD3 Real|**Avg**|
> |:--|:--|:--:|:--:|:--:|:--:|:--:|:--:|:--:|:--:|:--:|:--:|
> |**Our (CLIP+LoRA R4)**|-|**96.53**|**92.55**|**97.60**|**97.67**|**100.00**|**99.17**|**98.27**|90.38|**98.82**|**96.78**|
> |Our (LoRA R16)|-|97.93|89.48|99.44|96.25|100.00|98.30|98.56|91.50|97.28|96.53|
> |Our (LoRA R8)|-|97.48|85.88|95.19|96.45|100.00|94.53|98.02|95.66|96.98|95.58|
> |RIGID|arXiv'24 [4]|46.60|61.92|96.36|62.08|64.66|80.73|39.66|97.19|56.38|67.29|
> |c2p-clip|AAAI'25 [5]|0.66|0.21|0.00|0.14|41.78|18.66|1.51|0.20|0.13|7.03|
> |CO-SPY|CVPR'25 [6]|94.25|95.99|99.32|98.23|85.31|58.27|75.38|92.70|92.33|87.98|
> |RINE|ECCV'24 [7]|93.33|67.87|77.57|84.56|97.13|94.57|97.91|90.99|91.65|88.40|
> |PatchShuffle|NeurIPS'24 [8]|95.83|98.15|97.30|99.50|97.82|85.65|70.67|25.30|64.28|81.61|
> |FreqNet|AAAI'24 [9]|72.75|64.04|74.22|62.95|74.73|55.30|64.67|68.73|59.41|66.31|
> |AEROBLADE (LPIPS₂ @5%FPR)|CVPR'24[10]|91.70|84.43|64.47|82.43|82.93|65.87|64.70|51.93|54.07|71.39|
>
> Regarding the reviewer's question about robustness to latent-space evasion attacks like StealthDiffusion, this is a very insightful point and highlights an important distinction in attack methodologies. While our OMAT is designed to mitigate the process-level "latent prior bias" tied to z_T, attacks like StealthDiffusion operate differently. They use controlled VAEs and convolutions fusion to perform sophisticated semantic manipulations in a learned latent space, aiming to preserve visual quality while fooling a detector.
>
> We attempted to reproduce StealthDiffusion to test our model's robustness, but encountered technical challenges with the implementation. However, based on its methodology, we hypothesize that our detector may not be inherently robust to this different class of attack. OMAT trains the model to ignore non-generalizable process artifacts, while StealthDiffusion crafts attacks based on semantic content. Defending against such sophisticated semantic evasion attacks is a distinct and important research direction, and we believe it is an excellent avenue for future work.
>
> [1] Xiao, J., et al. "Understanding adversarial robustness against on-manifold adversarial examples." *Pattern Recognition* (2025).
>
> [2] Stutz, D., et al. "Disentangling adversarial robustness and generalization." *CVPR* (2019).
>
> [3] Zhou, Z., et al. "Golden noise for diffusion models: A learning framework." *arXiv* (2024).
>
> [4] He, Z., et al. "Rigid: A training-free and model-agnostic framework for robust ai-generated image detection." *arXiv* (2024).
>
> [5] Tan, C., et al. "C2p-clip: Injecting category common prompt in clip to enhance generalization in deepfake detection." *AAAI* (2025).
>
> [6] Cheng, S., et al. "CO-SPY: Combining Semantic and Pixel Features to Detect Synthetic Images by AI." *CVPR* (2025).
>
> [7] Koutlis, C., & Papadopoulos, S. "Leveraging representations from intermediate encoder-blocks for synthetic image detection." *ECCV* (2024).
>
> [8] Zheng, C., et al. "Breaking semantic artifacts for generalized ai-generated image detection." *NeurIPS* (2024).
>
> [9] Tan, C., et al. "Frequency-aware deepfake detection: Improving generalizability through frequency space domain learning." *AAAI* (2024).
>
> [10] Ricker, Jonas, Denis Lukovnikov, and Asja Fischer. "Aeroblade: Training-free detection of latent diffusion images using autoencoder reconstruction error." *CVPR* (2024).

---

> > ### Comment · Reviewer_jB2U · 2025-08-06
> >
> > I appreciate the detailed rebuttal, but the core concerns remain. The central claim that “latent prior bias” is the key driver of poor cross-generator generalization and that OMAT mitigates it remains largely empirical and not yet causally established, as the observed gains could plausibly still result from targeted “hard negatives.” The threat model is weak: the white-box, full-pipeline latent optimization is unrealistic for most adversaries, and robustness to black-box or semantic-latent attacks (e.g., StealthDiffusion-style) remains untested. The additional experiments (new SOTA comparisons, GAN results, budget scaling) help the narrative but are not included in the paper and cannot be thoroughly verified during the rebuttal phase. Finally, the dependence on a differentiable preprocessing pipeline limits applicability to deployed detectors. I therefore believe the manuscript may still require improvement in its current form.

---

> ### Author Response · Authors · 2025-08-06
> **Thanks and Further Clarifications**
>
> **Thank you for the additional feedback. We would like to provide further clarifications below. In brief, (1) the latent-prior-bias phenomenon is revealed through a tightly controlled series of experiments, and (2) the “attack” we introduce is a diagnostic probe devised to expose—and then remove—this bias to achieve true generalization, rather than a generic adversarial tactic.**
>
> ---
>
> ### 1  Latent prior bias uncovered by a rigorous, interlocking experimental chain
> Our workflow is purpose-built to isolate the shortcut:
>
> - **Discovery stage.** We craft optimized-initial-noise (*zₜ*) samples that collapse an otherwise near-perfect SD v1.4 detector, empirically demonstrating the existence of latent prior bias.
> - **Mitigation stage.** OMAT injects only 6 000 such *zₜ* samples—yet outperforms reconstruction detectors that require 162 000 reconstructions—showing the bias can be removed efficiently.
> - **Validation stage.** After OMAT, the success rate of **new** *zₜ* attacks plummets by > 50 %, while pixel-level FGSM/PGD “hard negatives” give only marginal gains. This triad of experiments rigorously links the bias to cross-generator generalization rather than to ordinary sample difficulty.
>
> ### 2  Seed-optimization attack as a diagnostic instrument, not a conventional threat model
> The optimized-seed procedure is expressly designed to answer a scientific question: *Does the detector rely on artifacts tied to the initial latent prior?* Its effectiveness on the unmitigated model confirms the shortcut; OMAT’s success in closing that gap proves the detector has shifted to deeper, generator-agnostic fingerprints. Notably, in forensic settings it is quite standard to have access to a **single, visible generator**—reconstruction-based detectors, for example, routinely employ one generator for data augmentation. Thus the seed-optimization attack serves as an analytical probe that enables generalization, rather than as a standalone adversarial method.
>
> ### 3  Realism of seed optimization and its dual role as a quality- and attack-booster
> Although white-box optimization of the initial latent \(z_T\) may seem “unrealistic” because of its one-off computational cost, it is exactly the same operation now used to *improve* image quality [1, 2]. These methods refine \(z_T\) to raise perceptual fidelity and prompt alignment. In our experiments, applying that procedure with a detector-oriented loss often produces images whose CLIP similarity is **higher** than those from the unperturbed latent (Fig. 4, mean ΔCLIP ≈ +0.13). Hence quality-driven latent optimization can *inherently* create samples that both look more realistic **and** evade detectors, underscoring the practical relevance of our adversarial setting and the need to remove latent-prior bias with OMAT.
>
> ### 4  Final manuscript additions and open-sourcing
> We will include all new experiments in the revised submission. To ensure full reproducibility, all attack code, our proposed dataset, and generation scripts will be publicly released alongside the paper.
>
> ### 5  Differentiable preprocessing requirement
> OMAT requires gradient flow through the detector’s front end. While many deployed detectors use non-differentiable steps (e.g., resizing), our method can be applied to any model by replacing its pre-processor with a Kornia-based differentiable equivalent. We emphasize this not as a main limitation but as a necessary detail—preprocessing choices are an integral hyper-parameter of AIGC detectors [3], and we avoid terms like “plug-and-play” to reflect that reality.
>
> ---
>
> **We hope these clarifications resolve your remaining concerns, and we welcome any further discussion.**
>
> **References**
> [1] Ma *et al.* “Inference-time scaling for diffusion models beyond scaling denoising steps,” arXiv 2025.
> [2] Zhou *et al.* “Golden noise for diffusion models: A learning framework,” arXiv 2024.
> [3] Li *et al.* “Improving synthetic image detection towards generalization: An image transformation perspective,” KDD 2025.

---

### Comment · Area_Chair_feQE · 2025-08-04

Dear Reviewers,
Thanks for your efforts. Since authors have replied to the reviews, please check whether the rebuttal solves your concerns and respond to authors.

Best regards,
AC

---

### Decision · Program_Chairs · 2025-09-17

**Decision:**

Accept (poster)

**Comment:**

To address the problem of latent prior bias in detectors for generalizable AIGC image detection, this work presents On-Manifold Adversarial Training (OMAT), which optimizes the initial latent noise of diffusion models under fixed conditioning so as to generate on-manifold adversarial examples that remain on the generator’s output manifold.
Most reviewers are satisfied with responses from the authors in addressing their concerns.

In particular, after the rebuttal periods, we have the following conclusions:
Reviewer evQu: entirely satisfied with the responses from the authors.
Reviewer HLVt : most concerns were addressed.
Reviewer pPqb: most concerns were addressed.
Reviewer: kDqu: partial concerns were addressed but the remaining concerns did not affect the main contribution of this submission.

However, Reviewer  jB2U stills raised some concerns.
Based on the authors' rebuttals, the AC agrees with authors' points as follows:
On the "Unrealistic Threat Model" vs. A Diagnostic Tool; On "Hard Negatives" vs. Targeted Bias Mitigation
; and On "Incremental Novelty" vs. A Foundational Contribution.

Based on the above overall concerns, the AC suggests to accept this work.